# TASK CONDITIONED STOCHASTIC SUBSAMPLING

## ABSTRACT

Deep Learning algorithms are designed to operate on huge volumes of high dimensional data such as images. In order to reduce the volume of data these algorithms must process, we propose a set-based two-stage end-to-end neural subsampling model that is jointly optimized with an *arbitrary* downstream task network such as a classifier. In the first stage, we efficiently subsample *candidate elements* using conditionally independent Bernoulli random variables, followed by conditionally dependent autoregressive subsampling of the candidate elements using Categorical random variables in the second stage. We apply our method to feature and instance selection and show that our method outperforms the relevant baselines under very low subsampling rates on many tasks including image classification, image reconstruction, function reconstruction and few-shot classification. Additionally, for nonparametric models such as Neural Processes that require to leverage whole training data at inference time, we show that our method enhances the scalability of these models. To ensure easy reproducibility, we provide source code in the **Supplementary Material**.

## 1 INTRODUCTION

Deep learning algorithms operate on large volume of high-dimensional dense inputs such as the pixels of an image (Deng et al., 2009; Krizhevsky et al., 2009; Liu et al., 2015). Training or evaluating a model with such data is computationally expensive and several works (Balın et al., 2019; Huijben et al., 2019; Yoon et al., 2018) have proposed subsampling techniques to subsample such dense inputs. Subsampling methods have the potential to drastically reduce the data acquisition effort and reduce the inference time of algorithms that operate on dense inputs. Additionally, subsampling techniques have found applications in medical research for the purpose of interpretation (Ribeiro et al., 2016).

However, these methods have a major drawback in that they are only applicable when each feature (such as a pixel) of the input is 1-dimensional. This restriction is imposed by the way the subsampling methods are designed: the input (e.g. an image), is flattened to a single vector and a model predicts a binary mask for each feature (e.g. pixels). This becomes problematic when we consider a 3-channel image. Flattening out the image results in ambiguities as to which pixels to select since channels are treated independently. For instance, in order to perform subsampling on CIFAR10 images, INVASE (Yoon et al., 2018) and DPS (Huijben et al., 2019) convert the images into single channel images (e.g. grey-scaled images) before subsampling pixels. In more extreme cases such as subsampling (different from instance-wise feature selection) of training instances, each feature is itself an image (each possibly multi-channel) and hence those subsampling techniques cannot be used. Finally, as we show in our experiments, most of these methods fail under extremely low subsampling rates and their performance is similar to random sampling in this setting.

In order to tackle these limitation, we propose to consider each feature or instance as an element of a set. We formulate the subsampling problem as selecting a subset of features, or instances that minimizes the performance degradation of an arbitrary model on an arbitrary task such as image classification, regression or instance subsampling for tasks such as few-shot classification. As a result, there are several advantages compared to the previous works. First, we can handle multi-dimensional feature, such as RGB pixel value, if we use set functions parameterized with expressive neural networks (Zaheer et al., 2017; Lee et al., 2018). Second, a subsampling model with set function can process arbitrary number of elements. Thus the model is robust to a wide range of subsampling rates at test time even when trained with fixed sampling rate. Lastly, the set-based formulation unifies the feature and instance subsampling tasks under a single framework.

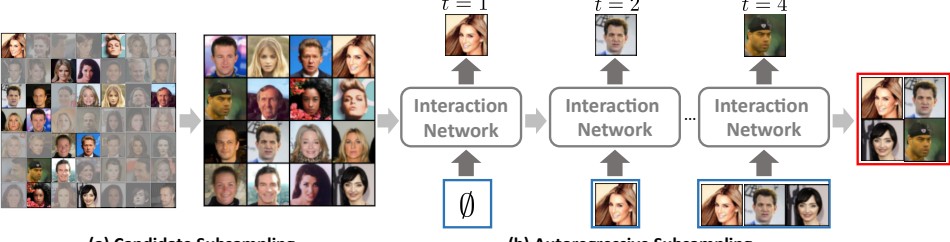

**Figure 1: Concept**: Two-stage stochastic subsampling process. **(a)** In the first stage, we screen out less important data points to construct the candidate subset. **(b)** In the second stage, we autoregressively select instances from the candidate subset.

However, it is prohibitively expensive to process all the set elements (e.g. all the pixels in an image) with expressive set function such as Set Transformer (Lee et al., 2018) due to the quadratic computational cost with respect to the number of elements. Hence we propose an efficient two-stage selection method. In the first stage, as shown in Fig. 1-(a), we learn the Bernoulli sampling rate for individual samples and efficiently screen out less important ones resulting in a subset which we call the *candidate set*. The second stage is more fine-grained and designed to select a smaller subset from the candidate set by considering the relative importance of the samples in the candidate set using a conditionally dependent Categorical distribution through an iterative procedure as shown in Fig. 1-(b). Once optimized, the resulting subsampling model can perform stochastic subsampling of a given input with linear time complexity. We call the resulting model *Stochastic SubSampling* (SSS) which is a general subsampling framework and can be applied to both *feature* and *instance* selection.

We validate SSS on multiple datasets and tasks such as 1D function regression, 2D image reconstruction and classification for both feature and instance selection. The experimental results show that SSS is able to subsample with minimal degradation on the target task performance, largely outperforming the relevant baselines. We summarize our contribution as follows.

- We reformulate the feature/instance subsampling problem by treating all the features/instances as members of a *set*. This allows us to apply set-based functions to the subsampling problem and extend its range of applicability.
- We propose a set-based two-stage stochastic subsampling method that learns to efficiently subsample a set with minimal performance degradation on a target task.
- We validate the efficacy and generality of our method on various datasets for feature selection in the input space (e.g. pixels) and instance selection from a dataset and show that it significantly outperforms the relevant baselines.

## 2 RELATED WORK

**Set Functions** Recently, extensive research efforts have been made in the area of set representation learning with the goal of obtaining order-invariant (or equivariant) and size-invariant representations. Many propose simple methods to obtain set representations by applying non-linear transformations to each element before a pooling layer (Ravanbakhsh et al., 2016; Qi et al., 2017b; Zaheer et al., 2017; Sannai et al., 2019). However, these models have limited expressive power. Yet, approaches such as Set Transformer (Lee et al., 2018) consider the pairwise interactions among set elements and hence can capture more complex statistics of set distributions. They often show better performance but require at least $O(n^2)$ time complexity.

**Feature Selection** Recent interest in deep learning based subsampling methods has produced many works mostly applied to feature selection. In Balın et al. (2019), continuous approximation of the Concrete Distribution (Maddison et al., 2016) is used for *global* feature selection where a fixed set of features are sampled across an entire dataset. In Chen et al. (2018), instance-wise feature selection is used for interpretation of deep learning models applied to medical data. Dovrat et al. (2019) propose learning to subsample by generating virtual points, then matching them back to the original input. Several works (Qi et al., 2017a;c; Li et al., 2018b; Eldar et al., 1997; Moenning & Dodgson, 2003) also propose *farthest point sampling*, which selects $k$ points from an input by ensuring that the selected samples are far from each other on a metric space. However our work is most similar with

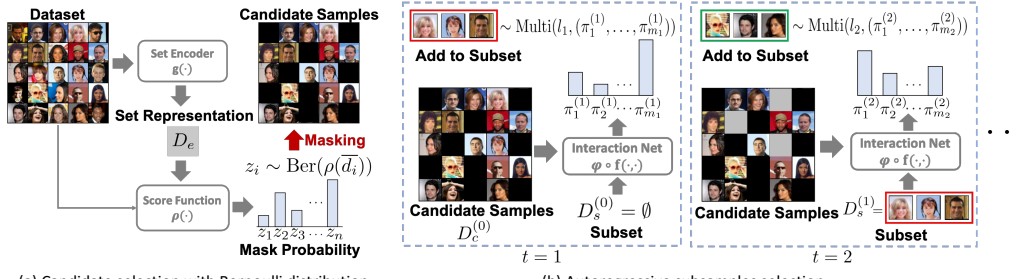

Figure 2: **Overview.** Two-stages of Stochastic Subsampling. **(a)** Construction of candidate subset $D_c$ with sampling mask for each instance from Bernoulli distribution. **(b)** Iteratively sampling elements from the candidate set with multinomial distribution to construct the subset $D_s$.

the recent works of Yoon et al. (2018) and Huijben et al. (2019) which learn a subsampling model conditioned on an given task. However these models have limitations both in terms of their range of applicability and poor performance under extremely low subsampling rates. Our method on the other hand is flexible, more general and applicable to a wide range of subsampling problems.

**Image Compression** Due to the huge demand for data transfer over the internet, some works attempt to compress images with minimal distortion. These models (Toderici et al., 2017; Rippel & Bourdev, 2017; Mentzer et al., 2018; Li et al., 2018a) typically consist of an encoder and decoder, where the encoder transforms the image with a compact matrix and the decoder reconstructs the image. These methods, while highly successful for the image compression problem, are less flexible than ours. Our model can be applied to any type of set structured data while the aforementioned models mainly work for images represented in tensor form. Furthermore, our method can be applied both at the instance and dataset level.

**Active Learning** Active learning aims to select data points for labeling given a small labeled set. This domain is different from ours since active learning does not consider the label information (for the selected data points) but our method does utilize label information. Also, our motivation is quite different. We focus on optimal subsampling conditioned on an arbitrary task and this greatly differs from the goal of active learning. Methods such as (Sener & Savarese, 2017; Coleman et al., 2019; Wei et al., 2015) all tackle the data selection problem in the active learning setting.

## 3 APPROACH

### 3.1 PRELIMINARIES

We consider a set $D = \{d_1, \ldots, d_n\}$ as an input with $D \sim p(\cdot)$ for some unknown data distribution, where individual $d_i$'s either represent a pair of *input* $x_i$ and *target* $y_i$ or a *feature* such as the pixel value of an image. We assume there exists a subset $D_s = \{s_1, \ldots, s_k\} \subset D$ such that $\ell(\cdot, D) \approx \ell(\cdot, D_s)$ for an arbitrarily defined loss function $\ell(\cdot, D)$ that we are interested in optimizing over the full set $D$ with $k \ll n$. In what follows, we present a method that learns the conditional distribution $p_\xi(D_s|D)$ of the subset $D_s$ via a two-stage subsampling method consisting of *candidate selection* and *autoregressive subset selection* as illustrated in Fig. 2. In general, we minimize the loss function $\mathbb{E}_{p(D)}[\mathbb{E}_{p_\xi(D_s|D)}[\ell(\cdot, D_s)]]$ with respect to $\xi$ the parameters of the subsampling model.

### 3.2 STOCHASTIC SUBSAMPLING

To select $D_s$, we need to model the pairwise interactions among the elements of $D$ and then choose a few representative elements in $D$ based on the relative sample importance computed using the interaction scores. However, when the cardinality of $D$ is large or its elements, $d_i$'s, are high dimensional, modeling pairwise interactions becomes computationally infeasible since we need to *compare every element in $D$ with all the other elements*. This computational bottleneck motivates the first stage of SSS of which the goal is to construct a smaller subset $D_c$, which we refer to as the *candidate set*, at a coarse level without considering pairwise interaction. We call the first stage *candidate selection* and the second stage, which is more fine-grained, *autoregressive subset selection*.

### 3.3 CANDIDATE SELECTION

We formulate the candidate selection problem as a random Bernoulli process where the parameters of the Beronulli distribution are conditioned on the set representation of $D$ and the individual elements $d_i \in D$. Specifically, we first encode the dataset $D$ to a single representation $D_e$ as follows:

$$D_e = \frac{1}{n} \sum_{i=1}^{n} g(d_i), \tag{1}$$

where $g$ is a neural network (Set Encoder in Fig. 2-(a)), which projects each element in $D$ independently to a lower dimension and $n = |D|$. This encoding scheme is similar to the one proposed in DeepSets (Zaheer et al., 2017) except that we do not perform message-passing between the set elements. We then concatenate every $d_i$ with $D_e$, denoted as $\overline{d_i}$. This ensures that each element of $D$ has a *global* view of all the other elements in the set at a coarse level. For every $d_i \in D$, we can sample a mask $z_i$ as follows:

$$p_\xi(z_i|d_i, D) = \text{Ber}(z_i|\rho(\overline{d_i})), \quad \rho(\overline{d_i}) = \frac{\sigma(h(\overline{d_i}))}{\sum_{j=1}^{n} \sigma(h(\overline{d_j}))} \tag{2}$$

where $h$ is a neural network that outputs the logits for the probability that $d_i$ is in the candidate set $D_c$ and $\sigma(\cdot)$ is the Sigmoid function, and Ber denotes the Bernoulli distribution. $z_i$ is a binary random variable where $z_i = 1$ indicates that $d_i$ is an element in $D_c$. We concatenate all $z_i$'s to obtain a single vector $Z = (z_1, \ldots, z_n)$. Note that we feed the output of $h$ into the Sigmoid function and normalize its output over all samples in $D$ to obtain a valid probability distribution and induce sparsity in the selected candidate set $D_c$. Since sampling from the Bernoulli distribution is not differentiable, during training, we use the continuous relaxations of the Bernoulli distribution (Maddison et al., 2016; Jang et al., 2016; Gal et al., 2017) to sample $z_i \sim \text{Ber}(\rho(\overline{d_i})$ for $d_i$ as shown in the black boxes in Fig. 2-(a). Although pairwise interactions are not considered in this stage, the ablation studies show that learning $p_\xi(z_i|d_i, D)$ leads to selecting highly informative samples compared to random selection.

**Constraining the size of** $D_c$ For computational efficiency, we want to restrict the size of $D_c$ to save computational cost when constructing $D_s$. Hence we introduce a sparse Bernoulli prior $r(Z)$ and minimize the KL divergence along with the target downstream task loss $\ell(\cdot, D_s)$ w.r.t $\xi$ as follows:

$$\mathbb{E}_{p(D)}\left[\mathbb{E}_{p_\xi(D_s|D)}[\ell(\cdot, D_s)] + \beta\text{KL}[p_\xi(Z|D)||r(Z)]\right], \text{ where } p_\xi(Z|D) = \prod_{i=1}^{n} p_\xi(z_i|d_i, D) \tag{3}$$

where $\beta > 0$ is a hyperparmeter used to control the sparsity level in $Z$.

### 3.4 AUTOREGRESSIVE SUBSET SELECTION

At this stage we have a set $D_c$ with $m = |D_c| \ll |D|$, which is small enough to perform fine-grained subset selection through pairwise modeling. To select a subset with $k$ elements from $D_c$, we requires $k$ iterative steps. As shown in Fig. 2-(b), at time step $t$, we have the subset $D_s^{(t-1)}$ constructed from the previous iteration with $D_s^{(0)} = \emptyset$ and $D_c^{(t)} = \{s_1^{(t)}, \ldots, s_{m_t}^{(t)}\} = D_c \setminus D_s^{(t-1)}$. Assuming we have a function $\varphi \circ f$ (Interaction Net in Fig 2-(b)) for modeling pairwise interactions between the elements of an input set, we autoregressively compute the interactions at time step $t$ as follows:

$$\tilde{\pi}^{(t)} = (\tilde{\pi}_1^{(t)}, \ldots, \tilde{\pi}_{m_t}^{(t)}) = \sigma(\varphi \circ f(D_c^{(t)}, D_s^{(t-1)})) \tag{4}$$

where $\sigma(\cdot)$ denotes the Sigmoid function and $\varphi \circ f$ is a composition of two neural networks, one that computes interaction scores between elements in $D_c^{(t)}$ and the other which outputs element-wise logits using the interaction scores. Further, $\tilde{\pi}^{(t)}$ is the vector of interaction scores for all elements in $D_c^{(t)}$ at the current time step $t$. Given $\tilde{\pi}_i^{(t)} > 0$ for all $i = 1, \ldots, m_t$, we can compute the probability of an element $s_i^{(t)}$ being selected from $D_c^{(t)}$ as:

$$p(s_i^{(t)}|D_c^{(t)}, D_s^{(t-1)}) = \pi_i^{(t)}, \quad \pi_i^{(t)} = \frac{\tilde{\pi}_i^{(t)}}{\sum_{j=1}^{m_t} \tilde{\pi}_j^{(t)}}, \quad m_t = |D_c^{(t)}|. \tag{5}$$

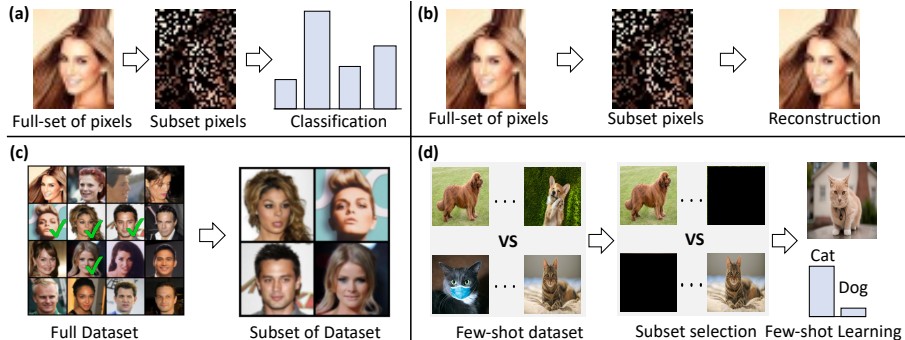

Figure 3: **Target Tasks:** **(a)** Feature selection for reconstruction. **(b)** Feature selection for prediction. **(c)**Instance selection for representative data points. **(d)** Instance selection for few-shot classification.

That is, we normalize $\tilde{\pi}^{(t)}$ over all the elements in $D_c^{(t)}$ at time step $t$ to obtain a valid probability distribution. The key to avoiding redundant elements in $D_s$ lies in the fact that for each element added to $D_s$, its selection is conditioned on both the candidate set $D_c^{(t)}$ and all the elements in the subset $D_s^{(t-1)}$ as described in Eq. 4 & 5. For the choice of the function $f$, we use a MultiHead Attention Block (MAB) (Lee et al., 2018) which we describe in detail in **Appendix H**.

With Eq. 5, we can sample an element $s^{(t)} \sim \text{Cat}(\pi_1^{(t)}, \ldots, \pi_{m_t}^{(t)})$ from the candidate subset $D_c^{(t)}$ and construct $D_s^{(t)} = D_s^{(t)} \cup \{s^{(t)}\}$, where Cat is Categorical distribution. During training, it can be expensive to sample $k$ times from the Categorical distribution since it involves computing Eq. 4 $k$ times. We remedy this by selecting $l$ elements from $D_c^{(t)}$ at once, which reduces the number of iterations to $k/l$ for selecting $k$ elements. We may sample $l$ elements from the multinomial distribution with probability $\pi^{(t)}$ without replacement. However, this sampling procedure is not differentiable, and hence it cannot be trained with backpropagation. Instead, we independently sample $l$ elements from the continuous relaxation of Categorical distributions (Maddison et al., 2016; Jang et al., 2016) using the same probabilities in Eq. 5 to approximate sampling from the multinomial distribution as shown in Fig. 2-(b). Since we want to simulate sampling without replacement, we discard all elements sampled more than once. This sampling procedure guarantees that we get at most $l$ elements at each iteration. A similar sampling procedure is adopted in previous works (Balın et al., 2019; Chen et al., 2018). We present this training algorithm in **Appendix C**.

**Inference Complexity of SSS.** The inference complexity of SSS depends heavily on the choice of the function $f$. Using MAB as $f$, the inference complexity of SSS is $O(n) + O(k^2 m/l)$ where $n, m, k$ correspond to $|D|, |D_c|$ and $|D_s|$ respectively.

## 3.5 TASKS

**Set Classification/Prediction** As shown in Fig. 3-(b), we train a neural network to predict a single target value $y_D$ for the subset $D_s$ of the given dataset $D$, where the set $D$ is a collection of the features from a single instance such as pixels of an image. For instance, the target is either the class of an image for classification or the statistics of the set for regression. Here, our goal is learning to select the most representative subset $D_s \subset D$ such that we can maximize the log likelihood $\log p_\theta(y_D|D_s)$ with computational efficiency. In order to achieve this goal, we jointly train the SSS model and the neural network which predicts the target value $y_D$ for $D_s$ to minimize the negative log-likelihood, the loss function $\ell(\cdot, D_s)$ described in Eq. 3 and enforce sparsity on $Z$, the selection masks for the candidate set, by minimizing the KL-divergence as follows:

$$\mathbb{E}_{p(D)}\left[\mathbb{E}_{p_\xi(D_s|D)}[-\log p_\theta(y_D|D_s)] + \beta \text{KL}[p_\xi(Z|D)||r(Z)]\right] \quad (6)$$

We provide experimental results in Section 4.4 and a corresponding graphical model in **Appendix E**.

**Set Reconstruction** Given a dataset $D = \{X, Y\}$ consisting of 2d coordinates $X = \{x_i\}_{i=1}^n$ and corresponding pixel values $Y = \{y_i\}_{i=1}^n$, we want to reconstruct all the RGB pixel values $y_i \in \mathbb{R}^3$ for each coordinate $x_i \in \mathbb{R}^2$ from the subset $D_s = \{X_s, Y_s\}$ with $X_s \subset X$ and $Y_s \subset Y$ as shown in Fig. 3-(a). We jointly train the SSS model and a neural network predicting pixel values to minimize

the loss function w.r.t $\theta$ and $\xi$ as follows:

$$\mathbb{E}_{p(D)}\left[\mathbb{E}_{p_\xi(D_s|D)}[-\log p_\theta(Y|X, D_s)] + \beta\text{KL}[p_\xi(Z|D)||r(Z)]\right] \tag{7}$$

We enforce sparsity on the subset $D_s$ by minimizing the KL-divergence between the mask probability $p_\xi(Z|D)$ and sparse prior $r(Z)$. Moreover, minimizing the negative log likelihood, which corresponds to $\ell(\cdot, D_s)$ in Eq. 3, ensures that the constructed $D_s$ is the most representative for the downstream tasks. We implement $p_\theta(Y|X, D_s)$ as an Attentive Neural Process (ANP) (Kim et al., 2019). The ANP takes $D_s$ as input and predicts a distribution of the elements in the original set $D$. It mimics the behaviour of a Gaussian Process but with reduced inference complexity. We present experimental results for this task in Section 4.3 and a corresponding graphical model depiction in **Appendix** E.

**Dataset Distillation: Instance Selection** In this task, we are given a collections of datasets $\mathcal{D} = \{D^{(1)}, \ldots, D^{(m)}\}$ with $D^{(i)} \cap D^{(j)} = \emptyset$ for $i \neq j$ and $D^{(i)} \overset{iid}{\sim} p(D)$, and the goal is to select the most representative subset $D_s$ with $k = |D_s| \ll |D|$ for each dataset $D = \{d_1, \ldots, d_n\} \in \mathcal{D}$, where $d_i$ is a data point uniformly sampled from the entire datasets $\mathcal{D}$. Using CelebA dataset as an illustrative example, shown in Fig. 3, $D$ consists of $n$ randomly sampled faces from the entire dataset and the task is to construct a subset, $D_s$, most representative of $D$.

In order to learn to select the subset $D_s$ from each $D \in \mathcal{D}$ with *unsupervised learning*, we jointly train the SSS model and a generative model such that the SSS model chooses the most representative subset so that the generative model can reconstruct all the images $d_i \in D$ from the subset. Naïvely, we can minimize the sum of negative log-likelihood $\sum_{d_i \in D} -\log p_\theta(d_i|D_s)$ for the loss function $\ell(\cdot, D_s)$ and KL divergence in Eq. 3. However, we find that the generative model outputs mean images for all $d_i$. To capture variations of different images, we introduce three latent variables $\alpha_i$, $c_i$, and $w_i$ which both depend on $d_i$. We provide graphical model illustration of this task in the **Appendix** E. Since it is intractable to compute the log likelihood $\log p_\theta(d_i|D_s)$ by marginalizing over all the latent variables, we derive the upper bound of the marginal likelihood using variational inference and plug the upper bound into the loss function $\ell(\cdot, D_s)$ in Eq. 3 as follows:

$$\mathbb{E}_{p(D)}\left[\mathbb{E}_{p_\xi(D_s|D)}\left[\sum_{d_i \in D}[\mathbb{E}_{q_\phi(w_i,c_i|d_i,D_s)}\left[-\log p_\theta(d_i|w_i, c_i)\right] + \text{KL}[q_\phi(w_i|d_i)||p_\psi(w_i)]\right.\right.$$
$$\left.\left. + \text{KL}[q_\phi(\alpha_i|d_i)||p_\psi(\alpha_i)] + \text{KL}[q_\phi(c_i|D_s, \alpha_i)||p_\psi(c_i)]]\right] + \beta\text{KL}[p_\xi(Z|D)||r(Z)]\right] \tag{8}$$

where $p_\psi(\cdot)$ are priors on their respective latent variables, $r(Z)$ is sparse prior over the mask for candidate set selection in SSS, $p_\theta(\cdot)$ is the decoder to reconstruct $d_i$, and all variational posteriors $q_\phi(\cdot)$ are parameterized with neural networks. All priors are chosen as the standard normal distribution.

In summary, we jointly train both the SSS and generative model to minimize the objective in Eq. 8 w.r.t $\theta, \phi$, and $\xi$ for all $D \in \mathcal{D}$ and leverage the optimized SSS to select a few representative instances of the dataset, which results in distilled dataset. We report all the experimental results in Section 4.5

**Dataset Distillation: Classification** Finally for the dataset distillation task, we consider the problem of selecting prototypes for few-shot classification as shown in Fig. 3-(d). We adopt Prototypical Networks (Snell et al., 2017) and deploy the SSS model for selecting representative prototypes from the support set for each class. We minimize the objective in Eq. 3, where we use the distance loss induced by the metric space from Prototypical Networks for the target task loss $\ell(\cdot, D_s)$, to jointly train the Prototypical Networks and SSS. Note that we use $D_s$, the subset of the support set, for computing loss and prediction. By learning to select the prototypes, we can remove outliers that would otherwise change the class prediction boundaries in the classification task where we need to predict the label $y_*$ for an unseen instance $x_*$. Experimental results for this task are in Section 4.5 and an accompanying graphical model description is provided in **Appendix** E.

## 4 EXPERIMENTS

Since SSS is a two-stage method, we perform extensive **Ablation in Appendix** I.

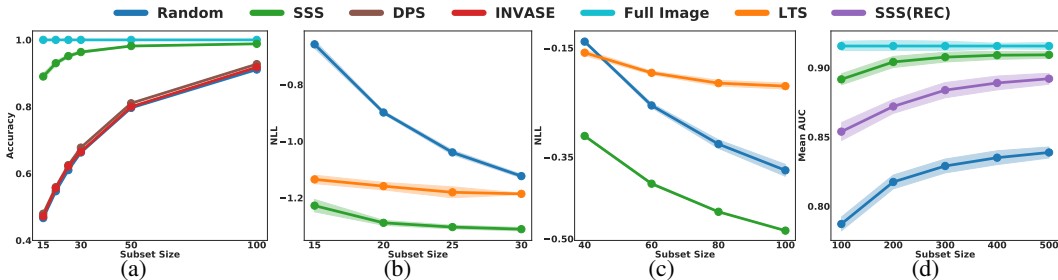

Figure 4: **(a)** MNIST Classification. **(b)** 1D Function Reconstruction. **(c)** Image Reconstruction on CelebA. **(d)** Attribute classification with feature selection on CelebA.

## 4.1 BASELINES

**Feature Selection for Classification** We compare SSS with the following models. 1) **Random Selection**: it randomly subsamples features. 2) **DPS** (Huijben et al., 2019): this model jointly optimizes the sampling parameters along with the parameters of a task model. 3) **INVASE** (Yoon et al., 2018): this model uses actor-critic (Peters & Schaal, 2008) to optimize the parameters of the sampling network, as well as a model that makes use of the full set of pixels in each image on the MNIST classification task (Section 4.2).

In Section 4.4, we perform attribute classification on the CelebA dataset using the selected features from a given image. We *cannot* apply DPS and INVASE to this experiment since each feature is a tuple of 3-d RGB pixels since flattening these features and applying DPS and INVASE results in ambiguities as to which features to select and which to ignore. For instance, applying these methods to a 3-channel image can result in some channels being selected while others are zeroed out. However this leads to the entire pixel being preserved and hence violate the subsampling objective.

**Feature Selection for Reconstruction** In Section 4.3 we compare SSS against the following: 1) **Random Selection**. 2) **LTS** (Dovrat et al., 2019): which is a model that learns to generate $k$ virtual elements which can be matched to elements in $D$ and optimized for the downstream task. We use LTS for both the function reconstruction and the image reconstruction tasks. DPS and INVASE are not applicable for these experiments since each feature is multi-dimensional. We also note that LTS is not applicable for the classification tasks since the virtual points generated by LTS cannot be converted back into image form to serve as input to an image classifier.

**Instance Selection** In Section 4.5, we compare SSS with 1) **k-Center-Greedy**: this algorithm iteratively selects elements in $D$ closest to a set of centroids and 2) **FPS**: this algorithm iteratively selects the most distant elements to a randomly initialized $D_s$ and **Random Selection** on the instance selection tasks. Here also, DPS, INVASE, and LTS are all inapplicable.

**Multiple Subsampling Rates** For all the neural network based baselines (DPS, INVASE, LTS), we train a separate model for every subsampling rate. For instance, to select 15, 20, 25, 30, 50 and 100 pixels from MNIST images in Section 4.2, we need to train 6 different models for DPS, INVASE and LTS each with the corresponding target subsampling rate. However for SSS, we train a single model and vary the sampling rate on each iteration. During evaluation, we use this single model for all the different sampling rates. We find that applying a similar training technique to the baselines result in drastic performance degradation. Thus the set formulation of SSS makes it generalize to varying subsampling rates at test time and offers train time efficiency.

## 4.2 FEATURE SELECTION FOR MNIST CLASSIFICATION

Given an MNIST image with 784 pixels, the task is to subsamlpe 15, 20, 25, 30, 50 and 100 pixels to be used as input to train and evaluate a classification model (CNN). Since images in the MNIST dataset have single channel, each feature is of dimension 1 and thus we can train a classifier with DPS or INVASE. We keep the pixels values for the selected pixels and set all the none selected pixels

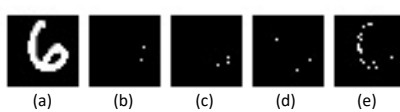

Figure 5: **(a)** Full-image, **(b)** Random, **(c)** DPS, **(d)** INVASE, **(e)** SSS. All models select 15 pixels of the original image.

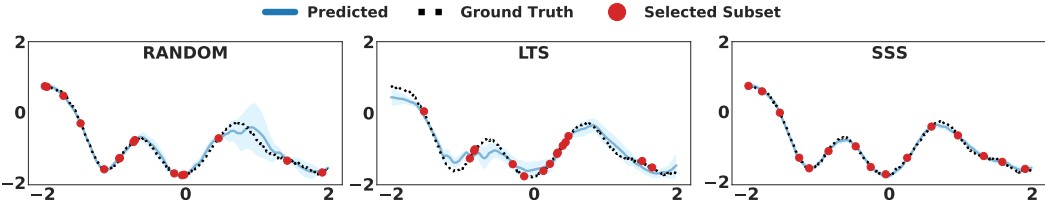

Figure 6: Visualization of 1D function reconstruction with **Random**, **LTS**, and **SSS**.

to zero. Note that we set subsampling rates to be much lower than the experimental setup from the papers (Yoon et al., 2018; Huijben et al., 2019).

As shown in Fig. 4a, SSS significantly outperforms all the baselines with large margin. It reaches 89% accuracy using only 15 pixels and shows better performance than the baselines with 100 pixels. Moreover under these low subsampling rates, the performance on the baselines are on par with random selection.

Lastly, we provide qualitative results in Fig. 5 where we visualize the selected 15 pixels by all the baselines and SSS. Again we find that SSS selects representative pixels (Fig. 5-(e)) so that the classifier can predict the correct label of the input image. However, all the baselines tend to select background black pixels, which are not informative for digit classification.

### 4.3 FEATURE SELECTION FOR REGRESSION

**Function Reconstruction** Suppose that we have a function $f : [a, b] \to \mathbb{R}$. We first construct a set of data points with $D = \{(x_1, y_1 = f(x_1)), \ldots, (x_n, y_n = f(x_n))\}$, where $(x_1, \ldots, x_n)$ are uniformly sampled from the interval $[a, b]$ and $f$ is Gaussian process. We sample $(y_1^{(i)}, \ldots, y_n^{(i)}) \overset{iid}{\sim} \mathcal{N}(\mathbf{0}, K_{XX} + \sigma_y^2 I_n)$ for $i = 1, \ldots, N$ where $K_{XX}$ is a squared-exponential kernel with the set of inputs $X = \{x_1, \ldots, x_n\}$ and $\sigma_y^2$ is variance for small likelihood noise. This leads to a collection of sets $(D^{(1)}, \ldots, D^{(N)})$. We train our model which consists of the subset selection model $p_\xi(D_s|D)$ and a task network $p_\theta(Y|X, D_s)$, which is Attentive Neural Process (ANP) (Kim et al., 2019), on this dataset and report the negative log-likelihood (NLL).

Fig. 4b shows the performance (NLL) of SSS compared to the baselines — Random Selection and LTS. As shown in Fig. 4b, SSS outperforms the baselines, verifying that the subset selection model $p_\xi(D_s|D)$ learns a meaningful distribution over subsets. We visualize a sample reconstructed function and the selected points by each models in Fig. 6. As shown in the rightmost figure (Fig. 6), SSS tends to pick out more elements (presented as red dots) in the drifting parts of the curve, which is reasonable since those are harder to reconstruct than the others. However, the other baselines sometimes fails to do that, which leads to inaccurate reconstructions.

**Image Reconstruction** Given an image, we learn to select a representative subset of pixels that best reconstructs the original image. Here, $x_i$ is the 2d pixel coordinates and $y_i \in \mathbb{R}^3$ is the RGB pixel value. We use an ANP to reconstruct the remaining pixels from the subset $D_s = \{x_{i_l}, y_{i_l}\}_{l=1}^k$ constructed by each subsampling model. We conduct the experiment on the CelebA dataset (Liu et al., 2018). Fig. 4c shows that our model significantly outperforms Random Selection and LTS in terms of NLL. We provide qualitative examples for the competing methods in **Appendix** F.2.

### 4.4 FEATURE SELECTION FOR CLASSIFICATION

In this subsection, we validate our model on the image classification task illustrated in Fig. 3-(b). The goal is to select a subset of pixels of an image and predict the label of the chosen subset. We jointly train the subsampling models and classifiers on the CelebA dataset, where each classifier performs binary classification for 40 attributes of a face. Note that only the selected pixels of an image by the subsampling models are used for prediction and the other pixel values are set to zeros.

We report the mean AUC score on all 40 attributes for varying sizes of $D_s$. Fig. 4d shows that using only 500 pixels ($\sim$1.3% of total pixels in an image), SSS achieves a mean AUC of 0.9093 (99.3% of the accuracy obtained with the full image). SSS achieves higher AUC score than Random Selection, showing the effectiveness of our subset selection method. We also include another baseline, namely **SSS-rec**. This is the SSS model trained for image reconstruction, but then later used for classification

Table 1: FID Score with varying the number of instances

| #Instances | 2 | 5 | 10 | 15 | 20 | 30 |
|---|---|---|---|---|---|---|
| K-Greedy | $8.8800 \pm 5.5857$ | $4.4306 \pm 1.3313$ | $4.2199 \pm 1.4214$ | $3.7160 \pm 1.1314$ | $3.2431 \pm 1.3881$ | $2.7554 \pm 0.8554$ |
| FPS | $6.5014 \pm 4.3502$ | $4.5098 \pm 2.3809$ | $3.0746 \pm 1.0979$ | $2.7458 \pm 0.6201$ | $2.7118 \pm 1.0410$ | $2.2943 \pm 0.8010$ |
| Random | $3.7309 \pm 1.1690$ | $1.1575 \pm 0.6532$ | $0.8970 \pm 0.4867$ | $0.3843 \pm 0.2171$ | $0.3877 \pm 0.1906$ | $0.1980 \pm 0.1080$ |
| **SSS** | $\mathbf{2.5307 \pm 1.3583}$ | $\mathbf{1.0186 \pm 0.1982}$ | $\mathbf{0.5922 \pm 0.3181}$ | $\mathbf{0.3331 \pm 0.1169}$ | $\mathbf{0.2381 \pm 0.1153}$ | $\mathbf{0.1679 \pm 0.0807}$ |

without any finetuning. Our model also outperforms this variant, showing the effectiveness of training with the target task. Note that we cannot apply LTS to this experiment because during training, the generated virtual points with LTS cannot be converted back to an image in matrix form due to the virtual coordinate, thus we cannot train the LTS model with CNN-based classification for this task.

**Inference Efficiency in Nonparametric models** In all the experiments where we used an ANP, we greatly improve the inference time complexity. By design, these models need to leverage the full training data at inference time. However by subsampling highly informative samples at low sampling rates (like in our experiments), these models must process very few elements at inference time. Similar gains can be obtained for models in the Neural Process family of models (Garnelo et al., 2018a;b).

## 4.5 DATASET DISTILLATION

**Instance Selection** The goal is to select only a few representative images from the given dataset as described in Section 3.5 and Fig. 3. We split the CelebA dataset into $m$ disjoint sets $\mathcal{D} = \{D^{(1)}, \ldots, D^{(m)}\}$ and jointly train SSS and the generative model to minimize the objective in Eq. 8 with $\mathcal{D}$. After training, we discard the generative model and leverage the subsampling model to choose a few representative images from the full CelebA dataset.

We evaluate the selected subset with the Fréchet Inception Distance (FID) Heusel et al. (2017), which measures similarity and diversity between two datasets and compare SSS to k-Center-Greedy, FPS and Random Selection. We report the experimental results in Table 1 where SSS achieves the lowest FID score for all selection sizes. Specifically, SSS outperforms all the baselines for selecting very few instances since SSS is able to model the interactions within the dataset and hence selects the most representative subset. Additionally, given that the dataset is highly imbalanced, k-Center-Greedy and FPS perform worst since by selecting extreme or similar elements in the given set cannot capture the true representation of the whole dataset. We provide selected images by SSS from the full dataset in **Appendix G**.

**Classification** In this task, we perform few-shot classification with the *mini*ImageNet dataset (Vinyals et al., 2016) where the models select 1, 2, or 5 instances from the support set with size 20. As shown in Table 2, we compare SSS against Random Selection, FPS, and k-Center-Greedy. SSS learns to select

Table 2: Accuracy on *mini*ImageNet

| #Instances | 1 | 2 | 5 |
|---|---|---|---|
| FPS | 0.432±0.005 | 0.501±0.002 | 0.598±0.000 |
| Random | 0.444±0.003 | 0.525±0.005 | 0.618±0.003 |
| K-Greedy | 0.290±0.006 | 0.413±0.005 | 0.570±0.002 |
| **SSS** | **0.475±0.006** | **0.545±0.011** | **0.625±0.006** |

more representative prototypes than the others especially for small $D_s$ where the choice of prototypes matters more. Notably, the K-Greedy method performs poorly for small subset sizes given that the model overfits to a few samples and does not generalize to unseen examples. We show samples of selected prototypes in **Appendix G.1**.

## 5 CONCLUSION

In this paper, we have proposed a stochastic subsampling method to reduce the size of an arbitrary dense input while preserving performance on a target task. Our subsampling method utilizes a Bernoulli mask to perform candidate selection, and a stack of Categorical distributions to iteratively select a representative subset from the candidate set. As a result, the selection process does take the dependencies of the input's elements into account. Hence, it can select a compact set that avoids samples with redundant information. By using the compact subset in place of the original set for a target task, we can save memory, communication and computational cost.

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

# A    ORGANIZATION

**Organization**   This supplementary file is organized as follows. We provide the full pseudo-code for the Greedy Training Algorithm and Fixed-size Subset Selection Algorithm and illustrate the generative process of each task using graphical models. We then provide qualitative results for the 1D function reconstruction task, qualitative results on the ablation studies and sample visualizations for the CelebA reconstruction task, Instance Selection and on Instance Selection as applied to few-shot classification on the *mini*ImageNet dataset. Finally we provide more details on specifications of models used in the experiments.

**Code.** We provide source code in the Supplementary Material.

---

**Algorithm 1** Greedy Training Algorithm

---

**Input**       $k$ (max subset size)
              $m$ (mini-batch size)
              $l$ (# elements selected at each iteration)
              $p(D)$ (distribution of sets)
              $\alpha$ (learning rate)
              $\ell(\cdot, D_s)$ (loss function)
**Output**    trained model with converged $\theta$ and $\phi$

1: Randomly initialize $\theta, \phi, \xi$.
2: **while** not converged **do**
3:       Sample $n$ sets $D^{(1)}, \dots, D^{(m)}$ from $p(D)$
4:       Sample $Z^{(j)} = \{z_1^{(j)}, \dots, z_n^{(j)}\} \sim p_\xi(Z|D^{(j)})$ for $j = 1 \dots m$
5:       Construct $D_c^{(j)} = \{d^{(j)} \in D^{(j)} : z^{(j)} = 1\}$ for $j = 1 \dots m$
6:       Sample integer $i \sim \text{Unif}[0, k - l]$
7:       $I^{(j)} \leftarrow$ random $i$-element subset of $D_c^{(j)}$ for $j = 1 \dots m$
8:       $Q^{(j)} \leftarrow$ select a $l$-element subset from $D_c^{(j)} \setminus I^{(j)}$ (with the auto-regressive model)
9:       $\theta \leftarrow \theta - \alpha \nabla_\theta \frac{1}{m} \sum_{j=1}^m \ell(\cdot, I^{(j)} \cup Q^{(j)})$
10:      $\phi \leftarrow \phi - \alpha \nabla_\phi \frac{1}{m} \sum_{j=1}^m \ell(\cdot, I^{(j)} \cup Q^{(j)})$
11:      $\xi \leftarrow \xi - \alpha \nabla_\xi \frac{1}{m} \sum_{j=1}^m \ell(\cdot, I^{(j)} \cup Q^{(j)})$
12: **end while**

---

# B    SOCIETAL IMPACT AND LIMITATION

Since our stochastic subsampling models learns to choose the most task relevant elements of a set, it does not explicitly handle any biases induced by the set where certain attributes are highly correlated with genders or races, and may select highly biased samples.

# C    GREEDY TRAINING ALGORITHM

In order to reduce the computational cost at training time, we use a greedy training algorithm with stochastic gradient descent as described in Algorithm 1. It selects only the top $l$ elements from the candidate set to train the auto-regressive model by minimizing the target loss on the selected samples and randomly selects the remaining $k - l$ elements from the candidate set. As a result, we do not have to run the auto-regressive model $k/l$ time during training, which significantly reduces the computational cost and shows reasonable performance.

# D    FIXED-SIZE SUBSET SELECTION

At test time, we run the fixed size subset selection algorithm to choose the most task relevant elements from the set $D$, as described in Algorithm 2. We do not use the greedy training algorithm. Instead, we autoregressively select $k$ elements from the candidate set $D_c$ as described in line 12 from Algorithm 2 to construct the representative subset $D_s$.

---

**Algorithm 2** Fixed Size Subsampling. $k$ is the required subset size. $l$ is the number of elements to select at each iteration. $D$ is the full input set and $D_s$ is the selected subset after running SSS.

1: **Input:** $k, l, D = \{d_1, \ldots, d_n\}$
2: **Output:** $D_s = \{s_1, \ldots, s_k\}$
3: **procedure** SSS$(k, l, D)$
4:     $D_e \leftarrow \frac{1}{n} \sum_{i=1}^{n} g(d_i), \overline{d_i} \leftarrow \text{Concat}(d_i, D_e)$
5:     $z_i \sim \text{Ber}(z_i | \rho(\overline{d_i})$ for $i = 1, \ldots, n$
6:     $D_c \leftarrow \{d_i \in D | z_i = 1 \text{ for } i = 1, \ldots, n\}$
7:     $D_s \leftarrow \emptyset$
8:     **for** $t = 1$ **to** $k/l$ **do**
9:         $D_s \leftarrow D_s \cup \text{AUTOSELECT}(l, D_s, D_c)$
10:     **end for**
11: **end procedure**
12: **procedure** AUTOSELECT$((l, D_s^{(t-1)}, D_c))$
13:     $D_c^{(t)} = \{w_1, \ldots, w_{m_t}\} \leftarrow D_c \setminus D_s^{(t)}$
14:     $\tilde{\pi}_i^{(t)} \leftarrow \sigma(\varphi \circ f(w_i, D_s^{(t-1)}))$
15:     $(\pi_1^{(t)}, \ldots, \pi_{m_t}^{(t)}) \leftarrow (\tilde{\pi}_1^{(t)}, \ldots, \tilde{\pi}_{m_t}^{(t)}) / \sum_{j=1}^{m_t} \tilde{\pi}_j^{(t)}$
16:     $Q \leftarrow \text{Select } l \text{ elements} \in D_c^{(t)} \text{ with } \pi^{(t)}$
17:     **return** $Q$
18: **end procedure**

---

## E    GRAPHICAL MODEL

In Figure 7, we illustrate the generative process using graphical models for each tasks — (a) feature selection for set reconstruction, (b) feature selection for prediction (c) Instance selection for representative data points and (d) instance selection for few-shot classification. Only the shaded circles denote observable variable and the others are latent variables.

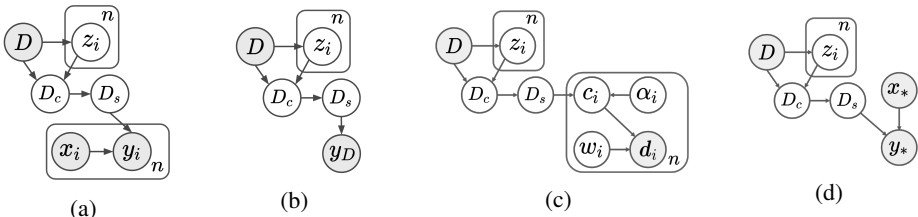

(a)          (b)          (c)          (d)

Figure 7: **Graphical Models:** **(a)** Feature selection for reconstruction. **(b)** Feature Selection for prediction task. **(c)** Instance selection for representative data points. **(d)** Instance selection for few-shot classification.

## F    INSTANCE SELECTION SAMPLES

In this section, we show more examples of our 1D and CelebA experiments on how the models select the set elements for the target task.

### F.1    1D FUNCTION - RECONSTRUCTION

Figure 8 shows the reconstruction samples of our model on the 1D function dataset, which is objectively better than that of Learning to Sample (LTS) or Random Subset (RS). Since RS selects the set elements randomly, it can leave out important part of the 1D curve leading to wrong reconstructions. LTS also selects insufficient amount of set elements in some parts of the curves, resulting in suboptimal reconstructions.

### F.2    CELEBA

Figure 9 shows samples of reconstructed images for varying selection number of pixels. Additionally in Figure 10, we show the selected pixels of our model for both the classification and reconstruction task. For the attribute classification task, the model tends to select pixels mainly from the face, since

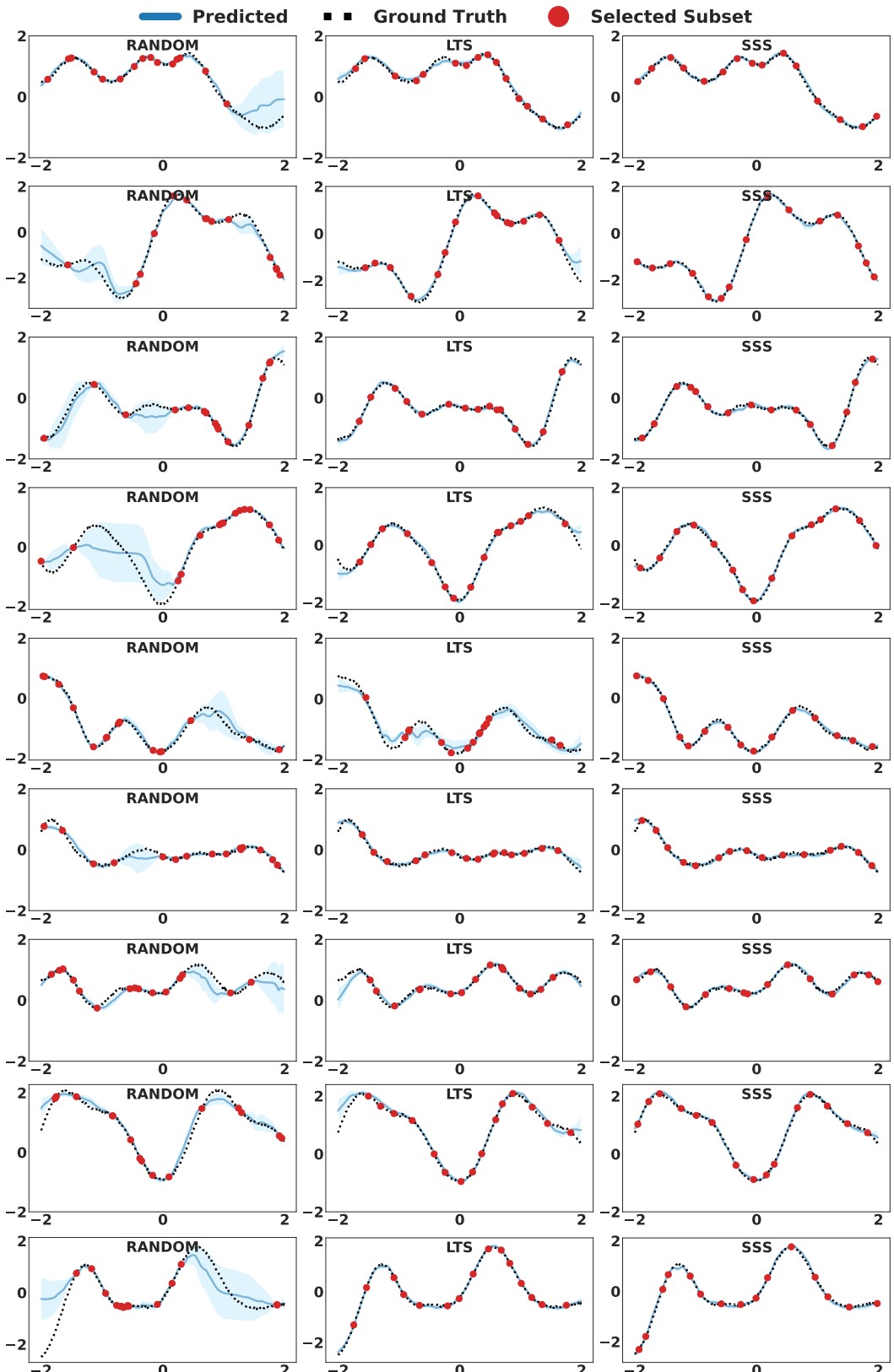

Figure 8: Visualization of 1D function reconstruction with three different subset selection models. Each method selects 15 out of 400 elements. As can be seen, SSS selects elements that result in better reconstructed functions.

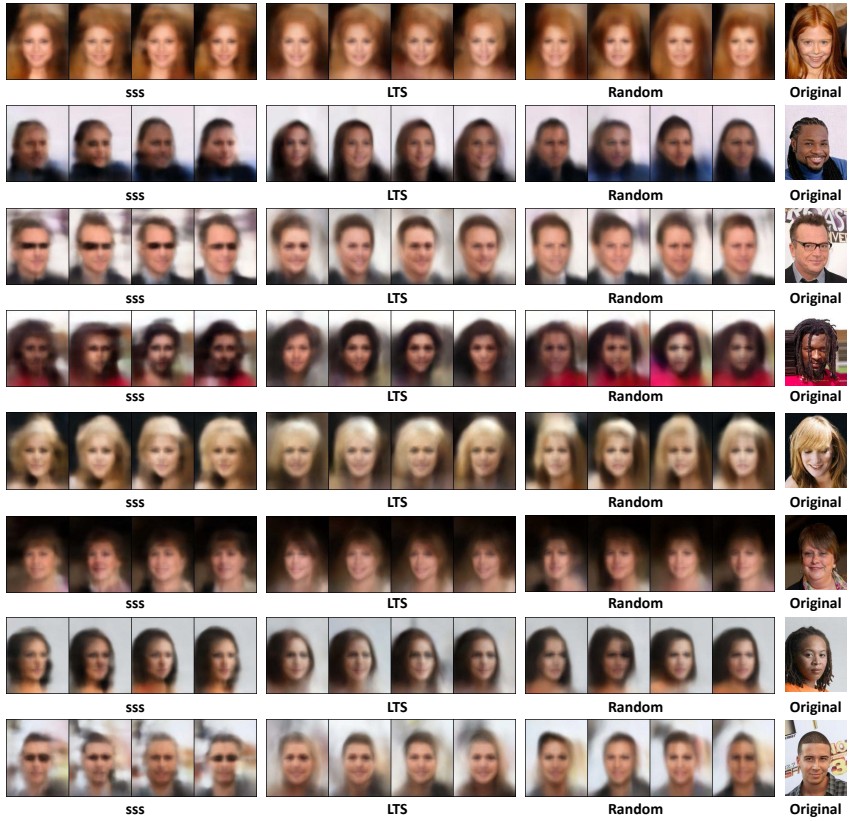

Figure 9: Visualization of reconstructed images for the CelebA dataset. Each model selects 40, 60, 80, and 100 pixels from a $218 \times 178$ image and reconstruct the full image using only the selected pixels.

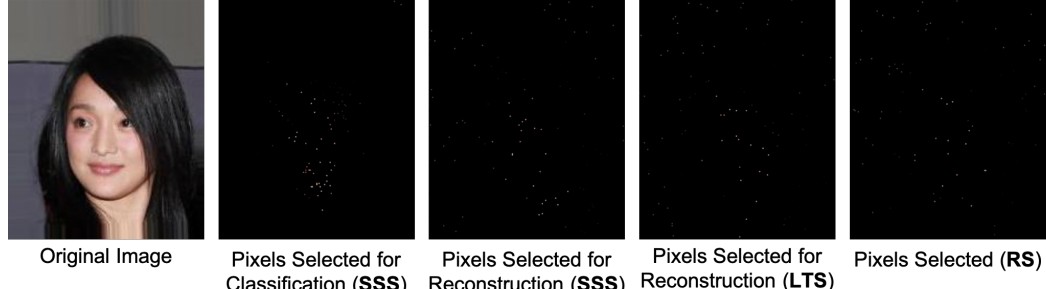

Figure 10: **Zoom-In for best view.** Selected pixels for different tasks on CelebA. As can be seen from the selected pixels, SSS adaptively selects different pixels for both reconstruction and classification. Pixels for reconstruction are more spread out to include the background since this contributes to the reconstruction loss. For classification, almost all the pixels are focused on the face since most of the attributes can be found there.

the task is to classify characteristics of the person. For reconstruction, the selected pixels are more evenly distributed, since the background also contributes significantly to the reconstruction loss.

# G    DATASET DISTILLATION: INSTANCE SELECTION

For the instance selection experiments, we construct a set by randomly sampling 200 face images from the full dataset. To evaluate the model, we create multiple such datasets and run the baselines(Random Sampling, K-Center Greedy and FPS) and SSS on the same datasets. The FID metric is then computed on the instances and averaged on all the randomly constructed datasets. For FPS, we use the open-source implementation in https://github.com/rusty1s/pytorch_cluster. Further, we provide qualitative results on a single dataset in Figure 11 where we show how our model picks 5 instances from the full set of 200 images face images.

## G.1    DATASET DISTILLATION: CLASSIFICATION

In Figure 12 we provide visualizations for the instance selection problem as applied to the few-shot classification task. Here, we go from a 20-shot to a 1-shot classification problem where the prototype is selected from the support using SSS. The selected subset is then used in place of the support set and used to classify new query instances.

# H    MODEL SPECIFICATIONS

In this section, we describe the main components of our Stochastic Subset Selection models — $g(d), \rho(\overline{d})$ and $\varphi \circ f(D_c^{(t)}, D_s^{(t-1)})$.

For all the experiments, we use $g$ feedforward neural network with ReLU to project each instance $d$ to lower dimension and average it to encode set representation $D_e$, following DeepSets (Zaheer et al., 2017).

We parameterize $\rho(\cdot)$ with a 3 layered feedforward neural network $h$ followed by sigmoid function as follows as described in Equation (2) from Section 3.3. For $\varphi \circ f$, we use conv-net to extract feature map for each instances in $D_c^{(t)}, D_s^{(t-1)}$ and feed it to set transformer (Lee et al., 2018) for set classification as follows:

$$f(D_c^{(t)}, D_s^{(t-1)}) = \text{MAB}(D_c^{(t)}, D_s^{(t-1)})$$
$$\text{MAB}(D_c^{(t)}, D_s^{(t-1)}) = \text{LayerNorm}(H + \text{rFF}(H)) \tag{9}$$
$$H = \text{LayerNorm}(D_c^{(t)} + \text{Multihead}(D_c^{(t)}, D_s^{(t-1)}, D_s^{(t-1)}))$$

where rFF is a row-wise feedfoward layer which processes each instance independently and Multihead denotes Multihead Attention (Vaswani et al., 2017) with each slot of Multihead$(\cdot, \cdot, \cdot)$ representing query, key, and value, respectively. For the other experiments, we use Deepsets for $f$. We use linear layer for $\varphi$ to output logits for each element in $D_s^{(t-1)}$.

## H.1    ATTENTION

We details on Attention and Multihead Attention here for completeness. For a more thorough exposition, we refer the reader to Vaswani et al. (2017) and Lee et al. (2018).

An attention module computes the following interactions using the dot product:

$$\text{Att}(Q, K, V; \omega) = \omega(QK^\top)V \tag{10}$$

where $Q \in \mathbb{R}^{n \times d_q}$ are the $n$ query vectors each with of dimension $d_q$. $K \in \mathbb{R}^{n_v \times d_q}$ and $V \in \mathbb{R}^{n_v \times d_q}$ are the keys and values respectively. Interactions are modelled through $QK^\top$ and $\omega$ is an activation function such as Softmax or Sigmoid.

Multihead attention projects $Q, K, V$ to $h$ different vectors each with $d_q^M, d_q^M, d_v^M$ dimensions and computes $h$ different attention modules according to the following:

$$\text{Multihead}(Q, K, V; \lambda, \omega) = \text{concat}(O_1, \ldots, O_h)W^O \tag{11}$$

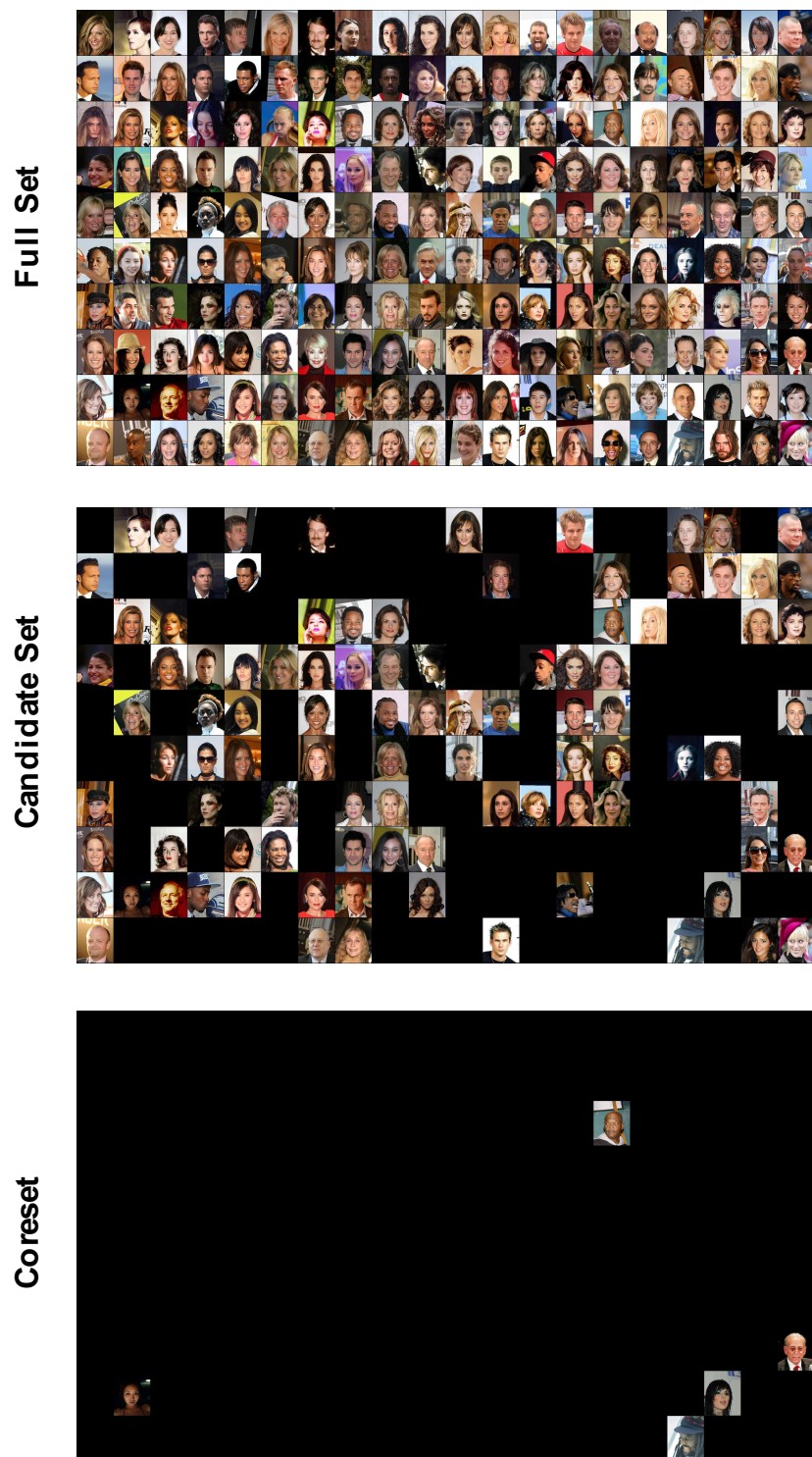

Figure 11: Visualization of a set with 200 images for instance selection. The two stage selection method in SSS is visualized as Candidate Set and coreset. A subset of size 5 is visualized.

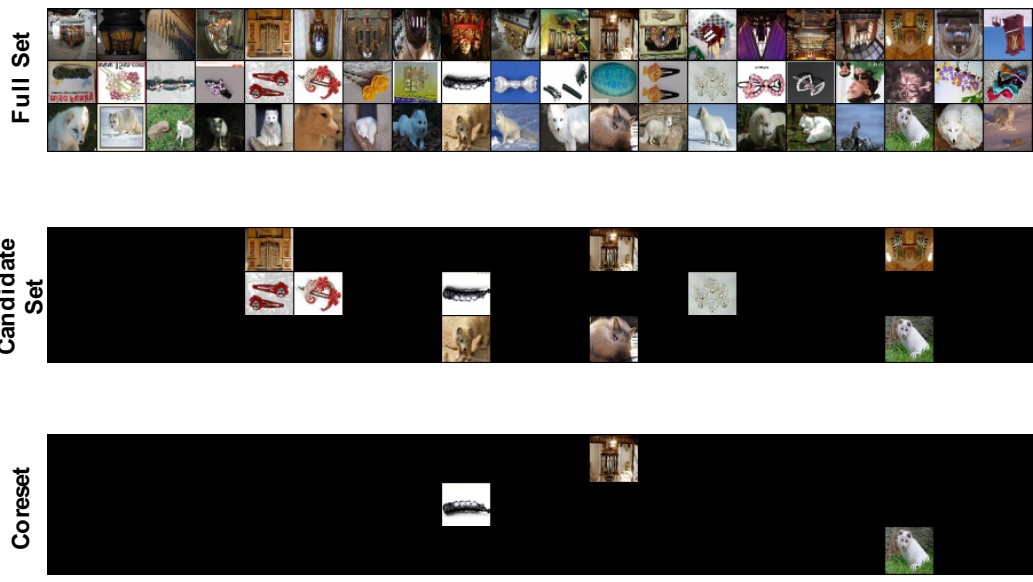

Figure 12: Sample visualization of prototype selection for the *mini*Imagenet dataset on the few-shot classification task. Each row represents a *set* that corresponds to the *support* from which a prototype is selected for the few-shot classification task.

where

$$O_j = \text{Att}(QW_j^Q, KW_j^K, VW_j^V; \omega_j) \tag{12}$$

The Multihead Attention module has learnable parameters $\lambda$ where $\lambda = \{W_j^Q, W_j^K, W_j^V\}_{j=1}^h$ and $W_j^Q, W_j^K \in \mathbb{R}^{d_q \times d_q^M}, W_j^V \in \mathbb{R}^{d_v \times d_v^M}$ and $W^O \in \mathbb{R}^{hd_v^M \times d}$. In all our experiments, we use Sigmoid as the activation function.

## I    ABLATION

We perform extensive ablation studies on the two stages of SSS using the function reconstruction task presented in Section 4.3. First we explore the contribution of the candidate selection and autoregressive subset selection stages. We then replace the candidate selection stage with random selection in the SSS model and compare it with the full SSS model.

**Candidate Selection Only** In Fig. 13, the model with only the candidate selection stage shows poor performance. This is because it is not always desirable to select only highly activating samples in the set without considering any dependencies among the others, which may leads to constructing a candidate set with redundant elements.

**Random Selection with Autoregressive Subset Selection** To show how much the candidate selection stage contributes to the performance of SSS, we replace it with random selection. As shown in Fig. 13, we find that while this model performs better than the model with only candidate selection, it performs worse than SSS and the autoregressive subset selection stage used alone. This shows that filtering elements with candidate selection helps the model to select more informative instances from the input set in the autoregressive subset selection stage than random selection.

**AutoRegressive Subset Selection Only** As shown in Fig. 13 we observe that the model with only autoreressive subset selection performs significantly better than the model with the candidate selection and the model with random selection followed by autoregressive subset selection. While this model performs well, it is not very practical due to the high computational cost when the size of the set becomes large. Additionally, processing lots of elements with the interaction model can lead

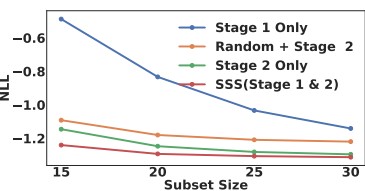

Figure 13: Ablation on SSS.

| Model | # Pixels | Storage | mAUC |
|---|---|---|---|
| Full Image | All 38804 | 114KB | 0.9157 |
| RS | 500 | 5KB | 0.8471 |
| SSS(rec) | 500 | 5KB | 0.8921 |
| SSS(MC) | 500 | 5*5KB | 0.9132 |
| SSS(ours) | 500 | 5KB | 0.9093 |

Table 3: CelebA Attributes Classification.

to flattened attention scores over the elements, which makes it difficult to distinguish important elements from the others. The candidate selection stage plays an important role of remedying this problem.

Generally, we find that SSS performs better than the variants considered here and has a better tradeoff between model performance and computational requirements.

**Stochasticity of SSS** Since our method is stochastic, the predictive distribution is $\mathbb{E}_{p_\xi(D_s|D)}[p_\theta(y_D|D_s)]$ for classification and we approximate it with Monte Carlo sampling. However in all experiments, we only report the result with one sampled subset, since it gives the best tradeoff in memory and computation. We compare it against another variant: **SSS-MC** which use 5 samples of subset for MC sampling and obtains a mean AUC of $91.32\%$. Note SSM-MC increases the computational cost (inference) and memory requirement up to 5 times. Our model SSS with a single subset sample achieves $90.93\%$ (Fig. 4d) accuracy, which is comparable to SSS-MC. The result justifies that our model SSS achieves good performance for target tasks with memory and computational efficiency. Storage comparisons for the selected pixels with these variants of SSS are presented in Table 3. Additionally, in Figure 14 we show how uncertainty of subset sampling method decreases as we increases subset size. For each subset size, we draw 5 different subsets from the subsampling model trained on MNIST classification task described in Section 4.2 and average prediction of the sampled subsets, which requires 5 forward pass of the model. After certain subset size, the model with single subset shows similar performance with the one with multiple draws of subsets.

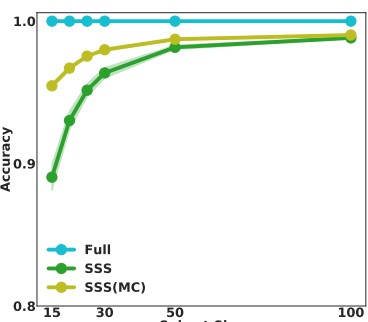

Figure 14: Accuracy with varying size of subset and the number of particles for MCMC.

**Cost of Running SSS** We report the cost of running SSS for pixel subsampling, which has the largest set size (38804 pixels). In this experiment, we select 500 pixels in total with $l = 20$, i.e., we select 20 pixels at once in the second stage described in Section 3.4. We measure the FLOPS and memory requirements for forward pass of SSS. We find that the computational cost of running SSS is 8.38 GMac (40% of the FLOPS for the full model which is 20 GMac) with 217.09k (compared to 958.85k for the full model) which shows that SSS is computationally cheap to run.

In the ablation studies, we investigated the contribution of the Candidate Selection stage (Stage 1) and the AutoRegressive subset selection stage (Stage 2). Additionally, we replaced Stage 1 with random selection in SSS and compare the performance of these three variants with SSS. In Figure 15, we provide qualitative results on the function reconstruction tasks for these 3 variants as well as SSS. As can be seen from the figure, using the Candidate Selection stage only to select elements generally focuses on the difficult sections of the curves ignoring the other parts which are relatively easier to reconstruct and this results in poorer overall reconstructions. The Autoregressive Subset Selection stage on the other hand selects elements that reconstruct the functions well. The downside to this model is that it requires a large amount of computation since we have to execute the interactions model for all elements in the input set and this renders this model impractical. On the other hand, the model that replaces the Candidate Selection stage in SSS with Random Selection performs better than using only the Candidate Selection stage only but results in function reconstruction that a sub-par compared to the Autoregressive Stage only and SSS. This is because using random selection in place of the first stage can ignore elements from certain parts of the function when we randomly sample the candidate

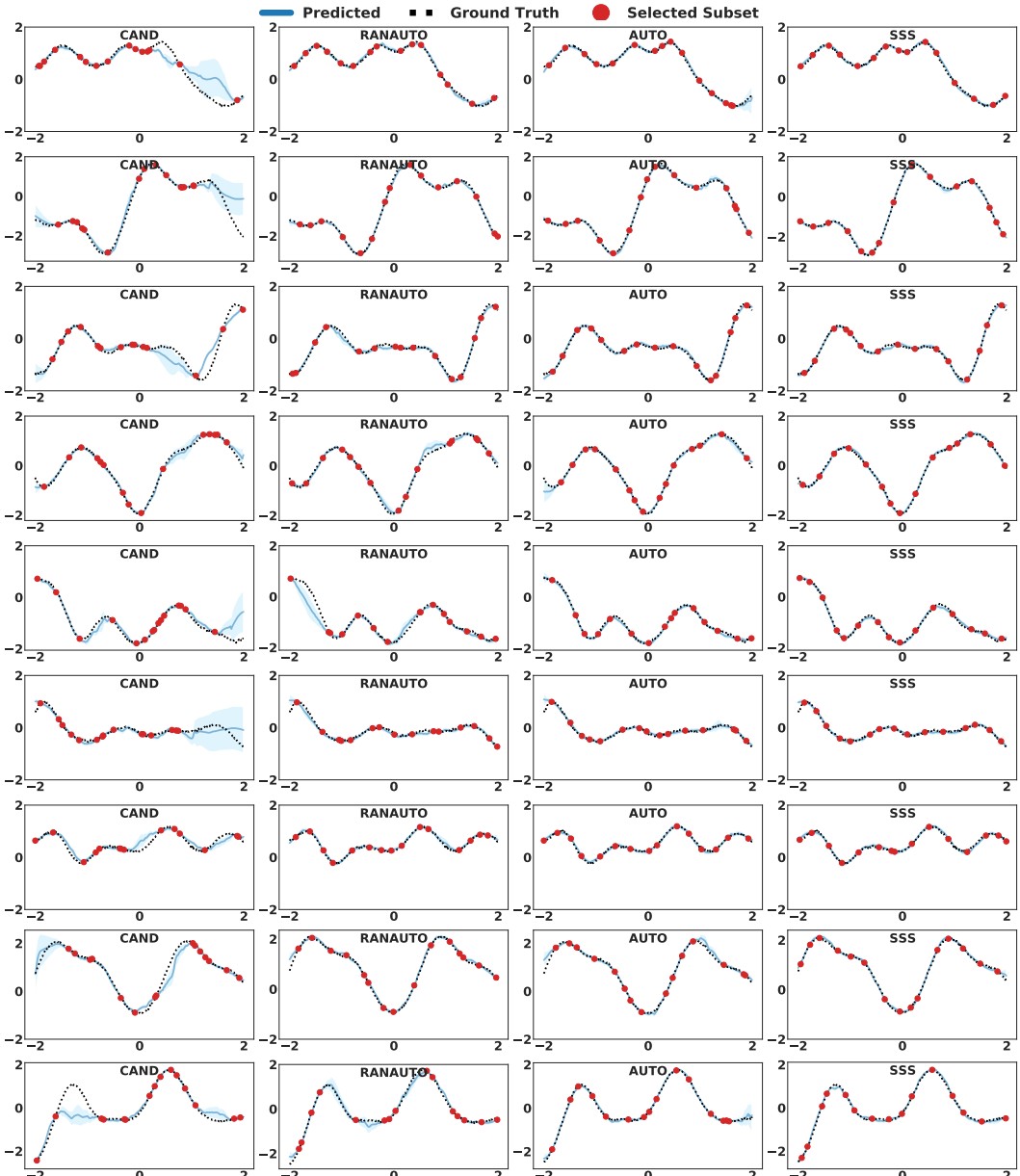

Figure 15: **Ablation:** Visualization of 1D function reconstruction. In the ablation studies, we compare the first stage (**CAND**) and the second stage (**AUTO**) with **SSS**. Additionally, we replace the Candidate Selection stage (Stage 1) in SSS with random selection (**RANAUTO**) and compare the performance of these models. As can be seen from the visualized reconstructed outputs, the combination of the Candidate Seletion stage with the Autoregressive stage results in the best subset selection for the reconstruction task.

set and hence these parts cannot be reconstructed since the second stage cannot select elements from those regions. This behaviour can be seen in some of the functions in the second column of Figure 15. In conclusion, SSS offers a good trade-off between computation and performance and results in better reconstructions since both the Candidate Selection stage and AutoRegressive Subset Selection stages work together to select the most relevant elements for the task.

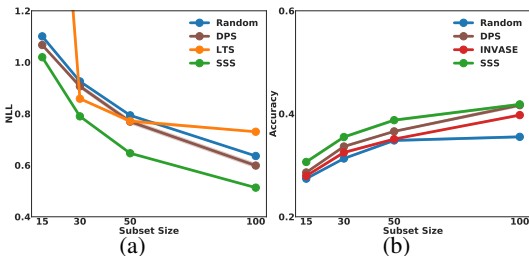

Figure 16: **(a)** CIFAR10 Reconstruction. **(b)** CIFAR10 Classification.

## J DISCUSSION OF LTS PERFORMANCE IN FIGURE 4C

For the image reconstruction task in Figure 4c, the LTS model generates virtual points for each pixel (RGB values and two coordinate values). In our experiments, we find that the generated coordinates values by LTS are imprecise compared to sampling directly from the original pixel space (as done in SSS and Random subsampling). This makes matching the virtual points with the original pixel values extremely difficult. Such inaccurate coordinate values result in poor performance as the subsampling rate increases even compared to random subsampling as depicted in Figure 4c. We observe similar pattern in the CIFAR10 reconstruction task presented in Section K and Figure 16a.

The need to predict the two coordinate values for the virtual points is a necessity only because the ANP model used for the reconstruction task requires the context points which consist of RGB values and their corresponding coordinates in the image space. Interestingly in the function reconstruction task in Section 4.3, the point matching stage in LTS is fairly easy and hence the LTS model shows better performance than random subsampling.

## K EXPERIMENTS ON CIFAR10 DATASET

We provide further experimental results on the CIFAR10 dataset in which we subsample pixels for both image reconstruction and image classification. In order to compare with both Huijben et al. (2019) and Yoon et al. (2018), we convert all the images to grayscale, following Huijben et al. (2019) (see Section 4.3).

The experimental result of the CIFAR10 reconstruction task is presented in Figure 16a, where for the same subsampling rates, SSS outperforms the competing baselines (DPS, INVASE, LTS and Random Sampling) in terms of the negative log-likelihood. For this task, we use ANP model as we have done in CelebA and function reconstruction tasks in Section 4.3. For this task, we do not compare with INVASE since INVASE requires two copies of the reconstruction model and requires more GPU memory.

In Figure 16b, we present the results for the CIFAR10 classification task. Again we observe that for the same subsampling rate, SSS performs better than DPS, INVASE and Random Sampling and the performance of the same classification model trained on the full input image is $0.70 \pm 0.02$. Note that we cannot compare with LTS for the same reasons given in Section 4.1.

