# OpenReview forum: "Task Conditioned Stochastic Subsampling"
_ICLR.cc/2022/Conference — ICLR 2022 Submitted_

### Official Review · Reviewer_zeQG · 2021-11-02

**Correctness:** 3
**Technical Novelty And Significance:** 3
**Empirical Novelty And Significance:** 3
**Recommendation:** 8
**Confidence:** 3

**Main Review:**

Strength:
- The paper writing is good, and the presentation is very clear.
- The idea of SSS is interesting, and its set-based nature allows it generalizes to various tasks.
- Extensive experiments have been conducted to support the claim that SSS is advantageous over other baselines.

Weaknesses:
-  In Figure 4 (c): for the task of image reconstruction on CelebA, I observe that LTS performs even worse than random subsampling, which is counter intuitive and is inconsistent with other tests, e.g., Fig 4 (b). Discussions need to be added here.
- In Section 3.5, I would suggest re-order the paragraphs so that they match the order of the illustrations in Figure 3 to increase the readability. Now it starts with Fig 3 (b), then (a), (c), (d). Why not (a), (b), (c) and (d)? Similar things could be improved in Section 4 when discussing each tasks and the experiments.

-Typo:
- In the third to last line of the paragraph after Eq. (2), there is a right parenthesis missed for "Ber(\rho(\bar d_i)"



**Summary Of The Paper:**

The paper proposes a Stochastic SubSampling method (SSS) that is helpful in reducing the size of input dataset while preserving reasonable performance on a target task. SSS is two-stage and set-based, which can be jointly optimized with arbitrary downstream tasks. Extensive experiments verify that SSS do perform well in the sense of efficiency and generalization a variety of tasks and datasets.

**Summary Of The Review:**

The paper writing is clear, and the proposed method is clearly advantageous over other baselines in terms of generalization and efficiency. Extensive experiments have been conducted to support the claim. I would vote for an acceptance of the paper.

---

> ### Author Response · Authors · 2021-11-19
> **Response to Reviewer zeQG**
>
> **[Q1]** In Figure 4 (c): for the task of image reconstruction on CelebA, I observe that LTS performs even worse than random subsampling, which is counter intuitive and is inconsistent with other tests, e.g., Fig 4 (b). Discussions need to be added here.
>
> - For the image reconstruction task in Fig 4(c), the LTS model generates virtual points for each pixel (RGB values and two coordinate values). In our experiments, we find that the generated coordinates values by LTS are imprecise compared to sampling directly from the original pixel space (as is done in SSS and Random subsampling). This makes matching the virtual points with the original pixel values extremely difficult. Such inaccurate  coordinate values result in poor performance as the subsampling rate increases even compared to random subsampling as depicted in Fig 4(c).
>
> - We would like to note that the need to predict the two coordinate values for the virtual points is a necessity only because the ANP model used for the reconstruction task requires the context points which consist of RGB values and their corresponding coordinates in the image space. Interestingly in the function reconstruction task in Section 4.3 (Fig 4(b)), the point matching stage in LTS is fairly easy and hence the LTS model shows better performance compared to random subsampling. We have included this discussion in the updated version of the paper (Appendix J).
> ---
> **[Q2]** In Section 3.5, I would suggest re-order the paragraphs so that they match the order of the illustrations in Figure 3 to increase the readability. Now it starts with Fig 3 (b), then (a), (c), (d). Why not (a), (b), (c) and (d)? Similar things could be improved in Section 4 when discussing each task and the experiments.
>
> - Thank you for the constructive suggestion. In the revised version of the papr, we have swapped the  figure 3(a) and figure 3(b) so that the order of figures match with the order of the task descriptions in section 3.5 and the order of corresponding experiments in section 4.
> ---
> **[Q3]** In the third to last line of the paragraph after Eq. (2), there is a right parenthesis missed for "Ber(\rho(\bar d_i)".
> - Thank you for pointing out the missing right parenthesis in Eq. (2). We have included it in the revised version of the paper.

---

> ### Author Response · Authors · 2021-11-26
> **A Gentle Reminder**
>
> Dear reviewer
>
> We sincerely appreciate your efforts in reviewing our paper, and your constructive comments. We have responded to your comments, faithfully reflected them in the revision, and provided additional experimental results. Could you please go over our responses and the revision since end of the final discussion phase is approaching? Please let us know there is anything else we need to clarify or provide.
>
> Thanks, authors.

---

### Official Review · Reviewer_oSsn · 2021-11-02

**Correctness:** 1
**Technical Novelty And Significance:** 2
**Empirical Novelty And Significance:** 4
**Recommendation:** 5
**Confidence:** 5

**Main Review:**

The major weakness of the paper is that it lacks theoretical justification for the proposed method. For example, it is unclear if the learned decision function (e.g., regression function and classification boundary) is consistent or not. Uncertainty of the learned decision function is unclear, either.

**Summary Of The Paper:**

This paper presents an interesting method for selecting subsamples or subfeatures used for deep learning. More precisely, the paper proposes to regularize the task loss by a Kullback-Leibler divergence between the selected samples/features and a sparse Bernoulli prior, and employs a neural network to learn the weights used for sample/feature selection.

**Summary Of The Review:**

This paper present an interesting method and promising results for subsample selection used in deep learning, while lacking theoretical justification for the proposed method.

---

> ### Author Response · Authors · 2021-11-19
> **Response to Reviewer oSsn**
>
> We would like to thank the Reviewer for the feedback. We respond to individual comments below.
>
> **[Q1]** The major weakness of the paper is that it lacks theoretical justification for the proposed method. For example, it is unclear if the learned decision function (e.g., regression function and classification boundary) is consistent or not.
>
> The contributions in this work are not theoretical analysis/justification in nature and we do not make any such claims in the paper.
>
> - We propose a novel two-stage subsampling method which is efficient and applicable to a wide range of tasks and empirically verify our method in accordance with previous subsampling works (DPS, LTS, INVASE, etc.). We extensively validate our method with empirical results.
>
> - Firstly, our proposed model SSS outperforms the most recent subsampling methods on various benchmark datasets. Secondly, the ablation studies (Appendix I) show that each component of our model is crucial for performance gain. Thirdly, thanks to the set-based formulation of the subsampling problem, we show that our proposed method can be  applicable to a wide range of tasks which baselines cannot tackle.
>
> - We want to emphasize that our work is not a **core-set** selection method in the framework of the works described in [1] where theoretical guarantees are necessary. **Subsampling** methods such as DPS, LTS, INVASE and ours are purely empirical works, and are presented as such, and hence we do not view the lack of theoretical guarantees as a limiting factor for these models and by extension ours.
> ---
>
> **[Q2]** Uncertainty of the learned decision function is unclear.
> - We respectfully disagree that the uncertainty of the learned decision function is unclear.
>
> - Firstly, we note that there is no uncertainty in the model parameters. The only other sources of uncertainty are in the selection of $D_s$ (due to the stochastic nature of SSS.) and uncertainty in the task model (such as an Attentive Neural Process) output. In Figure 6, we clearly show the uncertainty of the regressor, Attentive Neural Process.  As can be seen in Figure 6, for the same subsampling rate, $D_s$ obtained using SSS results in lower uncertainty (compared with the baselines) in the regressor output since SSS selects meaningful $D_s$ that results in better reconstruction.
>
> - Secondly, we state explicitly in Appendix I, that all the results presented in the paper utilize a single sample of $D_s$ using SSS for computation and efficiency reasons. Additionally in Appendix I, we discuss and provide experimental results (see Table 3 and Figure 14) that verify that: (a) multiple samples using SSS results in better performance and lower uncertainty of the task model output and (b) usage of a single $D_s$ sample (as was done for all the results reported in the paper) achieves performance that is comparable with the multiple sampled model and its uncertainty decreases as the subsampling rate increases. We further back the second point with new experimental results below where we evaluate the SSS model trained for MNIST classification over 5 MCMC samples (SSS_MCMC) and the model utilizing only a single sample (SSS). As can be seen from the results below, the performance of SSS matches SSS_MCMC as we increase the subsampling rate and its uncertainty decreases accordingly.
>
>
>
> ---
>
> **< Classification Accuracy on MNIST>**
>
>
> |   Method  |       15      |       20      |       25      |       30      |       50      |      100      |
> |--------|:-------------:|:-------------:|:-------------:|:-------------:|:-------------:|:-------------:|
> |    SSS    | $0.890\pm0.019$ | $0.930\pm 0.013$ | $0.951\pm 0.010$ | $0.964\pm 0.007$ | $0.982\pm 0.003$ | $0.988\pm 0.001$ |
> | SSS + MCMC | $0.955\pm 0.004$ | $0.967\pm 0.003$ | $0.975\pm 0.002$ | $0.980\pm 0.002$ | $0.987\pm 0.002$ | $0.990\pm0.001$ |
>
> ---
> # References
> [1] Feldman, Dan. "Introduction to Core-sets: an Updated Survey." arXiv preprint arXiv:2011.09384 (2020).

---

> ### Author Response · Authors · 2021-11-26
> **A Gentle Reminder**
>
> Dear reviewer
>
> We sincerely appreciate your efforts in reviewing our paper, and your constructive comments. We have responded to your comments, faithfully reflected them in the revision, and provided additional experimental results. Could you please go over our responses and the revision since end of the final discussion phase is approaching? Please let us know there is anything else we need to clarify or provide.
>
> Thanks, authors.

---

### Official Review · Reviewer_7mpS · 2021-11-06

**Correctness:** 2
**Technical Novelty And Significance:** 2
**Empirical Novelty And Significance:** 2
**Recommendation:** 5
**Confidence:** 3

**Main Review:**

Strengths:
1. The authors validate their approach on many different classification tasks.

1. The idea is simple and feasible.

1. The method offers performance improvements, although I'm skeptical about the empirical significance of the experiments.

Weaknesses:
1. The experiments are only on celebA faces and MNIST digits, both unchallenging datasets.

1. I do not think the numerical comparisons in Fig 4 are fair. The authors technically get access to the entire dataset and then choose what to show to the classifier, based on evaluations on different subsets. The performance gains over competing methods tell me that the baselines are most likely unfairly weaker.

1. I have concerns about the technical novelty of this work. It essentially boils down to sampling from a softmax distribution over features, then putting a sparsity prior on the distribution to make sure few features and data points are preserved.

**Summary Of The Paper:**

This paper proposes a data summarization technique that can be used for arbitrary downstream classification tasks. The authors show that their methods can be used to reduce the number of features required by a model while preserving accuracy. The authors additionally show that similar ideas can be used to extend this to selecting a subset of images in dataset that serves as a training dataset that provides good accuracy.

The algorithm proceeds by sampling features according to some softmax distribution defined by the classifier on each feature. The algorithm then learns to sample such that the accuracy of the classifier does not fall significantly when trained on a small subset of features. (The small subset of features are selected by adding a regularizer that rewards sparse selection of features)

**Summary Of The Review:**

Overall I am unconvinced by the technical novelty of the work and its implications. I do not necessarily buy the argument that existing feature selection techniques fail on RGB images. Additionally, I do not see the point of the experiments where a subset of the data is selected to train the classifier without sacrificing the accuracy -- each training technically requires the full dataset, even though the classifier gets to see only a subset, and training a new classifier would require the entire dataset to select a different sample.

---

> ### Author Response · Authors · 2021-11-19
> **Response to Reviewer 7mpS (4/4)**
>
> **[Q6]** Additionally, I do not see the point of the experiments where a subset of the data is selected to train the classifier without sacrificing the accuracy --- each training technically requires the full dataset, even though the classifier gets to see only a subset, and training a new classifier would require the entire dataset to select a different sample.
>
> - Firstly, we would like to clarify a factual misunderstanding. We do not subsample a subset of the **dataset** to train the classifier or regression models. In the image reconstruction task for instance,  pixels of each image form a **set** and the subsampling is performed only on the pixels (independently for each input image in the entire dataset). For the instance selection tasks in Section 4.5, a **set** consists of a few images, called support set in meta learning literature. We only subsample the instances of the support set, not the entire dataset.  We do not consider subset selection at the **dataset** level since this is not the problem domain we tackle in this work. We further elaborate it  and clearly distinguish subsampling methods (such as DPS, LTS, INVASE and ours) from core-set methods that aim to select a subset of a datatest for training.
>
> - Secondly we would like to emphasize again that our method is a **subsampling** method. This is very different from **core-set** selection methods [5] where the selected samples can be used to train a new classifier with lower computational and memory requirements. Subsampling methods (such as DPS, INVASE, LTS and ours) on the other hand do not aim to solve the same problem. The focus of subsampling methods is efficient inference with little degradation of  accuracy. That is, once the model has been trained, it subsamples a part of an input  to reduce the memory and computation requirements during evaluation. What we have proposed in this work (and to some extent by LTS, INVASE, DPS, etc) is to directly supervise the subsampling model with the target task objective. Importantly, subsampling methods have the potential to be used for explainability (such as in INVASE). In this light, the point of the experiments performed in our work and in the baselines become clear.
> ---
>
> # References
>
> [1] Garnelo, Marta, et al. "Conditional neural processes." International Conference on Machine Learning. PMLR, 2018.
>
> [2] Kim, Hyunjik, et al. "Attentive Neural Processes." International Conference on Learning Representations. 2019.
>
> [3] Yoon, Jaesik, Gautam Singh, and Sungjin Ahn. "Robustifying sequential neural processes." International Conference on Machine Learning. PMLR, 2020.
>
> [4] Lee, Juho, et al. "Bootstrapping neural processes." Advances in Neural Information Processing Systems 33 (2020).
>
> [5] Huijben, Iris AM, Bastiaan S. Veeling, and Ruud JG van Sloun. "Deep probabilistic subsampling for task-adaptive compressed sensing." International Conference on Learning Representations. 2019. (https://openreview.net/forum?id=SJeq9JBFvH)
>
> [6] Feldman, Dan. "Introduction to Core-sets: an Updated Survey." arXiv preprint arXiv:2011.09384 (2020).
>
> [7] https://github.com/jsyoon0823/INVASE
>
> [8] https://github.com/IamHuijben/Deep-Probabilistic-Subsampling
>
> [9] Snell, Jake, Kevin Swersky, and Richard S. Zemel. "Prototypical networks for few-shot learning." arXiv preprint arXiv:1703.05175 (2017).

---

> ### Author Response · Authors · 2021-11-19
> **Response to Reviewer 7mpS (3/4)**
>
> **[Q4]** It essentially boils down to sampling from a softmax distribution over features, then putting a sparsity prior on the distribution to make sure few features and data points are preserved.
>
> - We respectfully disagree with this summarization and we think this is an unfair characterization of our work and oversimplifies our method.  Firstly, we propose a novel two-stage subsampling model, which enables efficient processing of the large set of high-dimensional inputs such as high-resolution images (Sections 4.3 and 4.4).
>
> - In the first stage (Candidate Selection), we filter non-informative elements in the set, which significantly reduce the computational cost of fine-grained identification of $D_s$ in the second stage. In order to efficiently and effectively discriminate such elements, we capture global information efficiently by encoding the full input using a set encoder (Eq. 1) and use this as contextual information for coarse-grained subsampling. We showed in Appendix I that this first stage design offers performance gains.
>
> -  Also, only utilize the sparsity prior in the first stage, Candidate Selection, where the motivation is to drastically reduce the subsampling space for efficiency and is just a small part of the full model we present in Section 3.
>
> - In the second stage, our autoregressive sampler utilizes an expressive set encoding function which enables fine-grained subsampling. The encoder captures pairwise interaction between elements in the set, which helps the sampler to choose representative elements and maximize target task performance.
>
> ---
> **[Q5]** I do not necessarily buy the argument that existing feature selection techniques fail on RGB images.
>
> - We believe this comment is with regards to our explanation in Section 4.1 as to why it is infeasible to apply DPS and INVASE to multi-channel images (such as the experiments in FIgures 4(b-d) and the instance selection experiments in Section 4.5) in paragraph 2 of Section 4.
> We would like to point the Reviewer to Section 4.5 and Figure 5 in DPS [5], where the authors of [5] first need to convert all multi-channel images to grayscale before they apply subsampling method can to reconstruction of CIFAR10 images.
>
> - Again it is clear to see why this is necessary: their (both DPS and INVASE) formulation of the subsampling problem requires them  to operate directly on the inputs (this can also be verified by checking the **official implementation** of DPS and INVASE at [7] and [8]). They flatten a multi-dimensional input (e.g. an image) to a single vector and subsample a subset of elements from this vector. Multi-channel images particularly present a challenge since flattening it introduces ambiguities as to which pixels should be kept. For instance the R-channel of a pixel could be subsampled while the other two channels are not. In such cases it is unclear what to do.
>
> - In contrast, our set-based formulation of the subsampling problem naturally solves this issue. We cast an input into a set of elements with arbitrary dimensions and a subsampling model leans to choose a few representative elements from the set to minimize downstream task loss.
> ---

---

> ### Author Response · Authors · 2021-11-19
> **Response to Reviewer 7mpS (2/4)**
>
> **[Q2]** I do not think the numerical comparisons in Fig 4 are fair. The authors technically get access to the entire dataset and then choose what to show to the classifier, based on evaluations on different subsets. The performance gains over competing methods tell me that the baselines are most likely unfairly weaker.
>
> - This is again factually incorrect. All the baselines are evaluated exactly the same way our model is evaluated. Indeed the evaluation procedure is not a choice made by us. We follow the exact same evaluation procedure used by the competing baselines (DPS, INVASE, LTS).
>
> - We do not choose what to show to the classifier based on evaluations with different subsets. While our method is stochastic and allows us to perform multiple draws, we explicitly state in the Ablation study in Appendix I that all results presented in the paper utilize a single subset draw. This is consistent with the evaluation procedures of all the baselines. Additionally, we further demonstrate in Table 3 of Appendix I, that multiple draws do indeed provide better performance at the risk of increased computation and memory requirement hence our usage of a single draw in all our experiments.
>
> - Finally, our model does not get access to the entire dataset. This assertion is again factually incorrect. In the experiments for image reconstruction and image classification, the input to all the baselines and our model SSS is the full image, i.e. a set of pixel values **not** a set of training instances. All competing methods and ours then select a subset of the pixels in the image to be used for the downstream task (classification or reconstruction). In the Dataset Distillation experiments for both instance selection and classification, a datapoint consists of a collection of images, and here also, all the competing methods (including ours) take the full set of images as input.
>
> - In summary, **all** the competing methods (ours included) are evaluated exactly the same way and our method has no advantage over the baselines in this regard. Additionally, we would like to emphasize that both the **training and evaluation code are available** in the Supplementary material and it can be readily verified that our method has no unfair advantage over any of the baselines.
>
> ---
>
> **[Q3]** I have concerns about the technical novelty of this work.
>
> We believe our method has novelty and we recapitulate these points here.
>
> - We reformulate the subsampling problem by treating  features/instances as members of a set. This reformulation has a couple of advantages. Firstly, it allows us to apply set-based functions which extend the range of applicability of the proposed subsampling algorithm since almost any input can be represented as a set such as the pixels of an image. Secondly, this reformulation allows us to process inputs of arbitrary size both during training and testing. For instance, to evaluate DPS or INVASE on multiple subsampling rates at test time, one has to train a new model for each subsampling rate. Our set reformulation allows us to train a single model that is capable of handling arbitrary subsampling rates during both training and inference time. To the best of our knowledge, we are unaware of any work that takes the set approach that we present in this work.
>
> - We propose a set-based two-stage subsampling method that efficiently subsamples a set with minimal performance degradation (relative to training on the full input) on a target task. Our two-stage method is well motivated and can handle arbitrary data types and tasks which cannot be handled by the baselines such as LTS, DPS and INVASE. The candidate selection stage filters the input at a coarse level so that the autoregressive selection stage can be applied efficiently at a finer level of detail. Additionally, our autoregressive second stage itself has novelty for the subsampling task. Our employment of set-encoding functions in this stage allows us to efficiently capture information in $D_s$ while it is still being populated.
>
> - Lastly, our method has a wide range of applicability such as image classification (Section 4.2), image reconstruction (Section 4.3), function reconstruction (Section 4.3), support set selection for meta-learning (Section 4.5), and dataset distillation (Section 4.5). None of the baselines are applicable for all of these tasks.
>
> - To the best of our knowledge, we are the first to reformulate the subsampling problem with set-based formalism. Additionally, we are unaware of any subsampling method with a wide range of applicability as the method we present in this paper. Our 2-stage subsampling method is novel with each stage clearly motivated.
> ---

---

> ### Author Response · Authors · 2021-11-19
> **Response to Reviewer 7mpS (1/4)**
>
> We sincerely appreciate your constructive comments. We respond to the individual comments below:
>
> **[Q1]** The experiments are only on celebA faces and MNIST digits, both unchallenging datasets.
>
> - Firstly, this is factually incorrect. In the few-shot classification experiments described in **Section 4.5** and **Table 2** (see qualitative results in Figure 12 in the Appendix), we utilize the **mini-ImageNet** dataset under extreme subsampling rates (1, 2 and 5 instances for the support set). The mini-ImageNet dataset still remains as one of the challenging and most used benchmark dataset for few-shot classification.
>
> - Secondly, the CelebA dataset is a commonly used benchmark dataset for measuring the generalization performance of variants of neural processes [1,2,3,4]. In our experiments, we use an Attentive Neural Process [2] for the image reconstruction task and CelebA is the standard benchmark dataset for evaluating this model. Additionally we would like to add that selecting a subset of pixels for the CelebA reconstruction task is not a trivial problem as shown in the experimental results in Figure 4(c). Similar arguments can be made for the attribute classification results presented  in Figure 4(d) and Table 3 in the Appendix.
>
> - Lastly, our choice of the MNIST dataset (also used in DPS) is because the subsampling baselines, DPS and INVASE, cannot be applied for subsampling RGB colored images such as CIFAR10. Both these methods require the dimension of each element in the input (e.g. pixels) to be 1. Otherwise, flattening multi-channel inputs results in ambiguities as to which pixels to subsample since there is no guarantee that all channels belonging to a given pixel are selected or not, which we clearly described in the Introduction section of the main paper.
>
> - For instance, DPS (see Section 4.3 and Figure 5 of [5]) subsamples pixels of **grayscale** CIFAR10 images for reconstruction. The same issue arises when one tries to apply INVASE to multi-channel images. Our method however does not suffer from this limitation due to our set-based approach to the subsampling problem.  Hence the usage of MNIST here allows us to fairly compare our method with these two recent subsampling baselines.
>
> - Additionally, we have performed **classification** and **image reconstruction** on **grayscale CIFAR10** dataset. SSS still outperforms all the baselines (DPS, INVASE, LTS), which shows our subsampling method is applicable to a wide range of tasks.
> ---
>
> **< Classification Accuracy on grayscale CIFAR10>**
>
> | Method |       15      |       30      |       50      |      100      |
> |--------|:-------------:|:-------------:|:-------------:|:-------------:|
> | Random | $0.274\pm0.001$ | $0.313\pm 0.002$ | $0.348\pm0.004$ | $0.355\pm0.004$ |
> |  DPS   | $0.286\pm 0.001$ | $0.336\pm 0.004$ | $0.366\pm0.002$ | $0.417\pm0.006$ |
> | INVASE | $0.279\pm 0.003$ | $0.325\pm 0.004$ | $0.351\pm0.002$ | $0.397\pm0.004$ |
> |  SSS   | $\textbf{0.306}\pm 0.004$ | $\textbf{0.355}\pm0.004$ | $\textbf{0.388}\pm0.004$ | $\textbf{0.418}\pm 0.003$ |
>
>
> **< Image reconstruction negative log-likelihood on grayscale CIFAR10>**
>
> | Method |       15      |       30      |       50      |      100      |
> |--------|:-------------:|:-------------:|:-------------:|:-------------:|
> | Random | $1.101\pm0.00$ | $0.926\pm0.001$ | $0.794\pm 0.003$ | $0.636\pm0.000$ |
> | DPS    | $1.067\pm 0.002$ | $0.907\pm0.013$ | $0.769\pm0.016$ | $0.599\pm 0.017$ |
> | LTS | $2.742\pm 0.442$ | $0.859\pm0.002$ | $0.772\pm0.014$ | $0.730\pm0.000$ |
> | SSS    | $\textbf{1.020} \pm 0.002$ | $\textbf{0.790}\pm 0.002$ | $\textbf{0.647}\pm0.001$ | $\textbf{0.513}\pm0.001$ |

---

> ### Author Response · Authors · 2021-11-26
> **A Gentle Reminder**
>
> Dear reviewer
>
> We sincerely appreciate your efforts in reviewing our paper, and your constructive comments. We have responded to your comments, faithfully reflected them in the revision, and provided additional experimental results that you have requested. Could you please go over our responses and the revision since end of the final discussion phase is approaching? Please let us know there is anything else we need to clarify or provide.
>
> Thanks, authors.

---

### Official Review · Reviewer_Lpik · 2021-11-07

**Correctness:** 2
**Technical Novelty And Significance:** 2
**Empirical Novelty And Significance:** Not applicable
**Recommendation:** 3
**Confidence:** 3

**Main Review:**

The main contribution of this paper is to propose a 2-stage sampling method. The core method of each stage does not have much novelty. The main weakness of this paper is that the core details of the proposed method are missing in the main text, which makes it very difficult for readers to assess its true value.

Details:
1) p. 3, last paragraph of Sec. 3: It's not accurate that "active learning does not consider the label information." Action learning selects examples to be labeled, but which examples to select is based on the error/loss of the existing model, which in turn depends on the label information.
2) p. 3, Sec. 3.1: The authors state "we minimize the loss function..." Minimizing the loss function is to identify the subset $D_S$ or train the model for the downstream task? If it's the former, the previous description in Sec. 1 indicates that the authors would like to minimize the "performance degradation", which is the difference in losses from the original dataset $D$ and the subset $D_S$. If it's the latter, then why there're expectations here? Do you minimize over multiple draws or multiple selected subsets?
3) p. 4, Sec. 3.3: How do you train the neural network $h$?  It's not clear at all.  Based on Eq. (2), we have $E(\sum_{i=1}^{n} z_i) = 1$, which mean the effective sample size is 1? Also, you mentioned "sparse Bernoulli prior $r(Z)$". What does a sparse Bernoulli prior look like? Could you give an example?
4) p. 4, Sec. 3.4: For the neural networks $\varphi$ and $f$, how are they trained?
5) p. 6, Sec. 3.5: "Representativeness" is defined as allowing a generative model to reconstruct the other examples in the dataset. How do you use a generative model to reconstruct the remaining examples? There's no such guarantee by simply sampling from a VAE.


**Summary Of The Paper:**

This paper proposes a 2-stage sampling method that can find its use in either feature selection or instance (training examples) subsampling. At a high level, the authors frame the problem as a subset selection problem with the goal to minimize the performance difference for a downstream task between using the original set and the downsampled set. The proposed method consists of two stages: 1) select a subset (called candidate set) using Bernoulli variables as the binary mask, and the Bernoulli distribution's parameter depends on the corresponding example and the original set, 2) select elements from the candidate set iteratively to form the final selected subset, and the selection here is based on a softmax whose parameters depend on the interactions between the remaining candidate set and the selected subset so far. To support the merits of the proposed method, the authors provide experiment results on feature selection and dataset distillation tasks.

**Summary Of The Review:**

The authors try to pack lots of information into this paper, but the key details to support the usefulness of the proposed method are not fully explained. Without such details, it's hard for readers to appreciate its intellectual merits, and it's equally hard for readers to implement it in real applications. Therefore, this paper in its current form, doesn't seem to meet the bar of ICLR, and requires major improvements.

---

> ### Author Response · Authors · 2021-11-19
> **Response to Reviewer Lpik (3/3)**
>
> **[Q6]** If it's the latter, then why are there expectations here?
> - For large inputs such as high dimensional images, multiples subsets can represent the full input. That is, $D_s$ may not be unique. In order to explore a wide range of plausible subsets, we design a stochastic subsampling method so that we can draw multiple subsets $D_s\sim p_\xi(D_s|D)$. Therefore, we minimize the expected target task loss with respect to the subset $D_s$.
>
> ---
> **[Q7]** Do you minimize over multiple draws or multiple selected subsets?
> - With regards to multiple draws, our model is stochastic and hence we can perform multiple draws of subsets. However as we explain in the Ablation Study in Appendix I (Stochasticity of SSS), we only use a single draw in all our experiments. Additionally in Table 3, we show that while performing multiple draws (SSS(MC) in Table 3 and Figure 14) results in better performance, it has more computational and storage demands.
> ---
> **[Q8]** Based on Eq. (2), we have $E(\sum_{i=1}^{n}z_i) = 1$, which means the effective sample size is 1?
> - In Eq. (2), each $z_i$ is a random variable **independently drawn from the Bernoulli**
>  distribution and hence the $\mathbb{E}[\sum_{i=1}^n z_i] \neq 1$. In other words, the effective sample size is not 1. The effective sample size is controlled by the KL divergence between $p_\xi(Z|D)$ from candidate selection model and spares Bernoulli prior in Eq.(3).
> Note that the second stage does not require such KL term since we perform fixed-size subsampling of the candidate set in the second stage.
> ---
>
> **[Q9]** Also, you mentioned "sparse Bernoulli prior r(Z)". What does a sparse Bernoulli prior look like? Could you give an example?
> - A sparse Bernoulli prior is a Bernoulli random variable with a low $p$, the parameter for the Bernoulli distribution. In all our experiments, we set  $p=0.1$.
> ---
> **[Q10]** p. 4, Sec. 3.3: How do you train the neural network $h$? It's not clear at all. p. 4, Sec. 3.4: For the neural networks $\phi$ and $f$, how are they trained?
> - All the neural networks $h$, $\phi$ and $f$ are all  jointly trained  with the objective in Eq. (6), (7), or (8), depending on the downstream task. We have provided the detailed training procedure in Algorithm 1 (Appendix C). All the models are optimized exactly the same way a classifier or a regression model is trained.
> ---
> **[Q11]** p. 6, Sec. 3.5: "Representativeness" is defined as allowing a generative model to reconstruct the other examples in the dataset. How do you use a generative model to reconstruct the remaining examples? There's no such guarantee by simply sampling from a VAE.
>
> - Firstly, we **do not use** a VAE for the Dataset Distillation: Instance Selection task described in Section 3.5. We propose a new generative model in Section 3.5 whose graphical model is given in Figure 7(c) of Appendix E. In this model, we select the subset $D_s$ with the candidate selection stage and autoregressive selection stage as described in section 3.1 and 3.2. The role of $D_s$ is similar to the latent code of a VAE, which is a compact representation of input (full set for our model and an instance for VAE). Given the $D_s$, we reconstruct each $d_i $, element of the full set $D$, with the additional latent variables $\alpha_i, w_i, c_i$  for each $d_i$, which describes the characteristics of each element $d_i$.  The full model is presented in Section 3.5 together with the complete loss function.
>
> - Lastly, we can expect that model can reconstruct the full set $D$  from the subset $D_s$ since the proposed objective in Eq. (8) is the upper bound of the negative marginal log likelihood of the full set , $-\log p(D)$.
> ---
> # References
> [1] Sener, Ozan, and Silvio Savarese. "Active Learning for Convolutional Neural Networks: A Core-Set Approach." International Conference on Learning Representations. 2018.
>
> [2] Coleman, Cody, et al. "Selection via Proxy: Efficient Data Selection for Deep Learning." International Conference on Learning Representations. 2020.
>
> [3] Wei, Kai, Rishabh Iyer, and Jeff Bilmes. "Submodularity in data subset selection and active learning." International Conference on Machine Learning. PMLR, 2015.

---

> ### Author Response · Authors · 2021-11-19
> **Response to Reviewer Lpik (2/3)**
>
> **[Q3]** p. 3, last paragraph of Sec. 3: It's not accurate that "active learning does not consider the label information." Action learning selects examples to be labeled, but which examples to select is based on the error/loss of the existing model, which in turn depends on the label information.
> - In the active learning setting, one has to select a subset of **unlabeled** samples to be labeled using only a few labeled instances. Our method is not applicable in this setting since our subsampling method relies on the labels of individual instances (Section 3.5). This makes our subsampling method, and indeed those of LTS, DPS and INVASE, incomparable with active learning methods such as [1],[2] and [3]. We convey this difference more explicitly in the revised version.
> ---
>
>
> **[Q4]** p. 3, Sec. 3.1: The authors state "we minimize the loss function..." Minimizing the loss function is to identify the subset $D_s$ or train the model for the downstream task?
> - Identification of the subset $D_s$ and training of the model for the downstream task are **jointly optimized**. We do not train these separately. We minimize the loss on the downstream task with only $D_s$. The exact training algorithm is given in Algorithm 1 of Appendix C.
> ---
> **[Q5]** If it's the former, the previous description in Sec. 1 indicates that the authors would like to minimize the "performance degradation", which is the difference in losses from the original dataset  $D$ and the subset $D_s$.
> - In Sec. 1, we state that we would like to approximate $\ell(\cdot, D)$ with $\ell(\cdot, D_s)$. Performance degradation here is with respect to a model trained solely on $D$. While it is possible to directly minimize the difference between $\ell(\cdot, D)$ and $\ell(\cdot, D_s)$ (this is the approach taken by INVASE, one of the baselines), it would require two forward and backward passes which is computationally prohibitive. However, as we empirically show in all our experiments, directly minimizing $\ell(\cdot, D_s)$, results in a model that performs as well as a model trained on $D$ and is more computationally efficient.

---

> ### Author Response · Authors · 2021-11-19
> **Response to Reviewer Lpik (1/3)**
>
> We sincerely appreciate your constructive comments. We respond to the individual comments below:
>
> **[Q1]** The core method of each stage does not have much novelty.
>
> We believe our method is highly novel, and we recapitulate these points here.
>
> - We reformulate the subsampling problem by treating  features/instances as **members of a set**. This reformulation has a couple of advantages. Firstly, it allows us to apply set-based functions which extend the range of applicability of the proposed subsampling algorithm since almost any input can be represented as a set such as the pixels of an image. Secondly, this reformulation allows us to process inputs of arbitrary size both during training and testing. For instance, to evaluate DPS or INVASE on multiple subsampling rates at test time, one has to train a new model for each subsampling rate. Our set reformulation allows us to train a single model that is capable of handling arbitrary subsampling rates during both training and inference time. To the best of our knowledge, we are unaware of any work that takes the set approach that we present in this work.
>
> -  Secondly, we propose a novel **set-based two-stage subsampling** method that efficiently subsamples a set with minimal performance degradation (relative to training on the full input) on a target task. Our two-stage method is well motivated in that the  candidate selection stage filters the input at a coarse level so that the autoregressive selection stage can be applied efficiently at a finer level of detail. This two stage method enables efficient subsampling of large scale high dimensional data such as high resolution images (Sections 4.3 and 4.4).
>
> - Lastly, our method has **a wide range of applicability** such as image classification (Section 4.2), image reconstruction (Section 4.3), function reconstruction (Section 4.3), support set selection for meta-learning (Section 4.5), and dataset distillation (Section 4.5). None of the baselines are applicable for all of these tasks.
>
> - To the best of our knowledge, we are unaware of any subsampling method that reformulates the subsampling problem using set-based formalism. Additionally, we are unaware of any subsampling method with the range of applicability as the method we present in this paper. Our 2-stage subsampling method is novel with each stage clearly motivated
>
>
> ---
>
> **[Q2]** The main weakness of this paper is that the core details of the proposed method are missing in the main text, which makes it very difficult for readers to assess its true value.
>
> - In Section 3.2, we provide an overview of our proposed two-stage subsampling method. In Section 3.3, we provide the full details for the first stage, Candidate Selection. We follow this with a complete description of the second stage, Autoregressive Subset Selection in Section 3.4. In Algorithm 1 (Appendix C), we present the full greedy training algorithm discussed in Section 3.4 for training SSS and also present Algorithm 2 used for fixed-sized subsampling also described in Section 3.4.
>
> - Furthermore, we describe all the tasks we consider in Section 3.5 with illustrative figure 3 and further present 4 graphical models in  Appendix E  for each task.  It explains exactly how each model is implemented for each task. In Appendix H, we detail all the model specifications necessary to implement the presented method.
>
> - Moreover, we have included an extensive Ablation study in Appendix I which explores the contributions of each stage as well as explorations of other choices for the Candidate Selection stage such as random selection.
>
> - We further provide extensive qualitative results for all the tasks we consider both in the main paper and in the Appendix that demonstrate that the proposed method indeed subsamples meaningful subsets.
>
>  - Finally, we have included **source code** that implements the proposed subsampling method in the supplementary material.

---

> ### Author Response · Authors · 2021-11-26
> **A Gentle Reminder**
>
> Dear reviewer
>
> We sincerely appreciate your efforts in reviewing our paper, and your constructive comments. We have responded to your comments, faithfully reflected them in the revision, and provided additional experimental results. Could you please go over our responses and the revision since end of the final discussion phase is approaching? Please let us know there is anything else we need to clarify or provide.
>
> Thanks, authors.

---

### Author Response · Authors · 2021-11-19
**Summary of the Revision**

**Summary of the Revision**

We really appreciate all the reviewers for their constructive comments. We have responded to the individual comments from the reviewers below, and believe that we have successfully responded to most of them. We have included the discussions and results of the suggested experiments in the revision. Here we briefly summarize the updates we have made to the revision:

- We have included **additional experiments** --- classification and image reconstruction on gray-scale CIFAR10 datasets, as suggested by **Reviewer 7mps**

- We have **elaborated distinction** between subsampling and corset selection methods, as suggested by **Reviewer 7mpS**.

- We have **swapped the order** of figure 3(a) and (b), as suggested by **Reviewer zeQG**.

- We have further distinguished the problem settings we consider from active learning methods, as suggested by **Reviewer Lpik**

---
**< Classification Accuracy on grayscale CIFAR10>**

| Method |       15      |       30      |       50      |      100      |
|--------|:-------------:|:-------------:|:-------------:|:-------------:|
| Random | $0.274\pm0.001$ | $0.313\pm 0.002$ | $0.348\pm0.004$ | $0.355\pm0.004$ |
|  DPS   | $0.286\pm 0.001$ | $0.336\pm 0.004$ | $0.366\pm0.002$ | $0.417\pm0.006$ |
| INVASE | $0.279\pm 0.003$ | $0.325\pm 0.004$ | $0.351\pm0.002$ | $0.397\pm0.004$ |
|  SSS   | $\textbf{0.306}\pm 0.004$ | $\textbf{0.355}\pm0.004$ | $\textbf{0.388}\pm0.004$ | $\textbf{0.418}\pm 0.003$ |


**< Image reconstruction negative log-likelihood on grayscale CIFAR10>**

| Method |       15      |       30      |       50      |      100      |
|--------|:-------------:|:-------------:|:-------------:|:-------------:|
| Random | $1.101\pm0.00$ | $0.926\pm0.001$ | $0.794\pm 0.003$ | $0.636\pm0.000$ |
| DPS    | $1.067\pm 0.002$ | $0.907\pm0.013$ | $0.769\pm0.016$ | $0.599\pm 0.017$ |
| LTS | $2.742\pm 0.442$ | $0.859\pm0.002$ | $0.772\pm0.014$ | $0.730\pm0.000$ |
| SSS    | $\textbf{1.020} \pm 0.002$ | $\textbf{0.790}\pm 0.002$ | $\textbf{0.647}\pm0.001$ | $\textbf{0.513}\pm0.001$ |

---

### Author Response · Authors · 2021-11-23
**Dear Reviewers,**

Could you kindly go over our responses and the revisions we have made based on your suggestions since the interaction period comes to an end this Monday(22nd)? We have responded to your queries in detail and provided additional experimental results. We are grateful for your constructive feedback.

Sincerely, Authors.

---

### Decision · Program_Chairs · 2022-01-20

**Decision:**

Reject

**Comment:**

The paper proposes a set-based neural subsampling model that selects both features and instances using a two-stage model. The motivation is to allow for scaling to large datasets by first subsampling using an initial stage that does not model second-order interactions (which would require work quadratic in the dataset size to model), and then following up with a second more sophisticated pass that includes second-order interactions. Its results show an empirical improvement over previous methods in the case of very small subsample sizes.

The responses from the reviewers in discussion were varied—and often off-base for all reviewers—and as a result I took an even more deep look at this paper than usual. I think the variance in reviewer response is a symptom of the fact that the paper is somewhat confusingly written, and sometimes has parts that give the opposite impression to what the authors intend. For example, the author response says "we emphasize again that our method is a subsampling method. This is very different from core-set selection methods" but then Figures 11 and 12 explicitly label the subset produced by the algorithm a "coreset." There is general confusion as to what exactly is being subsampled (features or instances) and even what datasets were being evaluated on—it's not that the information is not there, but rather that it's easy to miss while reading the paper. We can see this happening where most of the reviewers were misunderstanding or making factual errors about the paper, and I can see how this happened by reading the paper.

The reviewers also shared some concerns about the baselines, and indeed some things about the baseline comparisons are confusing: for example, Figure 4 seems to report DPS having below 70% accuracy on MNIST while subsampling to a size of 25, but the original DPS paper (Huijben et al, ICLR 2020) in their Figure 2 reports a percent error of 6.6% (at Pixels removed: 96.8%, which I believe corresponds to keeping 25 pixels as 28*28*(1-0.968) = 25). This does seem to back up the reviewer's speculation that "the baselines are most likely unfairly weaker." The presentation of the results should make this sort of thing more clear (if the setup isn't the same as DPS's MNIST setup, how does it differ? if the setup is the same, as seems to be the case the way the paper is presently written, why are the result accuracies so different from what is published in the DPS paper?).

The reviewer comment about theory is not one I count against the submission. Although it is certainly true that this paper would be greatly strengthened by some theoretical backing, it is also part of a line of work that eschews theory—so we cannot reasonably disqualify it for doing so.

To sum: although the technical contributions of this paper do seem to be significant, I expect that if the paper is published as presently written, it will confuse ICLR readers just as it has confused our reviewers. This leads me to lean against accepting this paper at this time.